# Nanocarrier mediated delivery of insecticides into tarsi enhances stink bug mortality

Sandeep Sharma [1], Thomas M. Perring[2], Su-Ji Jeon [1], Huazhang Huang [3], Wen Xu[3], Emir Islamovic[3], Bhaskar Sharma [1], Ysabel Milton Giraldo [2] & Juan Pablo Giraldo [1] ✉

Current delivery practices for insecticide active ingredients are inefficient with only a fraction reaching their intended target. Herein, we developed carbon dot based nanocarriers with molecular baskets (γ-cyclodextrin) that enhance the delivery of active ingredients into insects (southern green stink bugs, *Nezara viridula* L.) via their tarsal pores. *Nezara viridula* feeds on leguminous plants worldwide and is a primary pest of soybeans. After two days of exposure, most of the nanocarriers and their active ingredient cargo (>85%) remained on the soybean leaf surface, rendering them available to the insects. The nanocarriers enter stink bugs through their tarsi, enhancing the delivery of a fluorescent chemical cargo by 2.6 times. The insecticide active ingredient nanoformulation (10 ppm) was 25% more effective in controlling the stink bugs than the active ingredient alone. Styletectomy experiments indicated that the improved active ingredient efficacy was due to the nanoformulation entering through the insect tarsal pores, consistent with fluorescent chemical cargo assays. This new nanopesticide approach offers efficient active ingredient delivery and improved integrated pest management for a more sustainable agriculture.

The United Nations established zero hunger by 2030 as a sustainable development goal to achieve food security[1,2]. Stink bugs (Hemiptera: Pentatomidae) are a major pest of food crops affecting over 60 different crops worldwide, causing annual losses ($60 million) that may surpass those caused by other insects[3]. Soybean (*Glycine max* L.) is an important crop accounting for 53% of the world's oilseed production[4] that is severely impacted by the southern green stink bug (*Nezara viridula* L.)[5]. This stink bug causes damage to soybean foliage and beans by using their piercing-sucking stylets to inject digestive enzymes into the plant tissue and sucking out plant fluids. This results in loss of turgidity, delayed maturation, stunted growth, and undersized or aborted seeds[6]. The primary strategy for managing *N. viridula* relies on the use of synthetic insecticides, including systemic insecticides (e.g., neonicotinoids) and non-systemic insecticides (e.g., pyrethroids)[7]. The main exposure routes of stink bugs to insecticides are from contact and oral exposure, depending on where the insecticide is located. Although insecticide formulations are effective in controlling insects, only a small fraction (1–25%) of applied insecticide active ingredient (AI) reaches their intended target insects[8,9]. Furthermore, AIs can be susceptible to degradation by bacteria inhabiting the insect gut, leading to diminished insecticide effectiveness[10]. All these factors often necessitate the use of high rates and multiple applications of insecticides causing unintended non-target and environmental impacts[11,12]. To address this challenge, a transition towards the development of targeted insecticide delivery systems is increasingly needed to improve the efficacy, safety, and sustainability of pest management.

With an expanding global population and the looming threat of climate change impacting crop production, nanotechnology has flourished as a valuable tool to improve sustainable agriculture and

[1]Department of Botany and Plant Sciences, University of California, Riverside, CA 92521, USA. [2]Department of Entomology, University of California, Riverside, CA 92521, USA. [3]BASF corporation, 26 Davis Drive, Research Triangle Park, NC 27709-3528, USA. ✉e-mail: juanpablo.giraldo@ucr.edu

provide food security[13,14]. Engineered nanomaterials (ENMs) hold great promise in the development of new technologies and strategies for crop pest and pathogen management[15–19]. Their unique physico-chemical properties have been explored for controlled and targeted delivery applications in plants[20–22], presenting a tremendous potential to improve insecticide efficiency while reducing their environmental impact[23–25]. Ongoing research is dedicated to optimizing ENMs delivery efficiency by manipulating their size, charge, surface area, and polarity[26,27]. Notably, recent studies have investigated how the physi-cochemical properties of ENMs influence their interaction with and penetration through, the insect cuticle. Hemocytotoxic peptide-conjugated nanodiamonds when applied topically to the cuticle sur-face of a beetle pest (*Tenebrio molitor* L.), were able to migrate through cuticle pores because of their smaller size (<10 nm) than the pore canals (6–65 nm)[28]. Likewise, polyanhydride ENMs (150 nm) dis-tributed more uniformly and in higher concentrations (~2.2-fold) across the exterior of mosquito cuticle compared to microparticles (~2 μm) and entered the internal organs via migration through cuti-cular sclerite junctions[29]. These amphiphilic polyanhydride ENMs were found to be associated with internal tissues to a higher degree than hydrophobic ENMs. Overall, these studies indicated that ENMs can enter insects through various parts of the insect cuticle.

Recently, ENMs such as polymer, graphene oxide, mesoporous silica, and clay have enhanced the protection of insecticide AIs from degradation and reduced loss in the environment. Mesoporous silica ENMs loaded with the insecticide acetamiprid exhibited similar aphid mortality as compared to only acetamiprid. However, the nano-formulation offered advantages by enhancing the wettability and adhesion ability of acetamiprid while demonstrating no effect on the germination rate of *Vicia faba* L. seeds[30]. Recently, a clay-based pyrethrin formulation at a half dose showed similar mortality of *Galleria mellonella* L. larvae compared to a full dose of pyrethrin alone. This was attributed to the effectiveness of clay nanotubes to protect the pyrethrins from photodegradation and hydrolysis while improving their aqueous solubility[31]. Similarly, zein polymer-based avermectin nanoformulation caused enhanced mortality (27%) of *Plutella xylostella* L. moths by improving the avermectin photo-stability under UV irradiation, in comparison to avermectin emulsi-fiable concentrates[24]. Graphene oxide-based chlorpyrifos AI nanoformulation enhanced mortality (35%) of *Pieris rapae* L. larvae compared to chlorpyrifos emulsifiable concentrate due to their high adhesion ability on the leaf surface under simulated rainy conditions[23]. In all of these studies, the nanoformulations do not directly increase insect mortality by improving the *delivery* of insecticide AIs. Instead, their primary function is to safeguard against the loss and degradation of AIs under various environmental condi-tions and thereby, conserve AI availability to insects. Although, some studies demonstrated the penetration of ENMs across the aphid cuticle, enhancing the uptake of RNA-based insecticides[32,33], these studies delivered the insecticide through the notum (dorsal exos-keleton of the thorax), which is not in direct contact with the leaf surface as is the insect distal leg segments (tarsi). Furthermore, with the exception of the aphids, these studies have been conducted on defoliators, in which the larvae directly consume the leaf mixed with the nanoformulations, allowing direct entry of insecticide AIs into their bodies. *Nezara viridula*, like other stink bugs, is a piercing-sucking insect that feeds by penetrating plants and sucking sap, rather than consuming leaves[34]. Nanotechnology-based approaches have not been implemented for insecticide delivery to stink bugs. Previous studies have identified the presence of pores in the cuticle of *N. viridula* tarsi, which are used to secrete pheromones or other physiological substances[35,36]. In this study, we aimed to determine if insecticide active ingredient delivery could be mediated by nano-carriers to the stink bugs via tarsal pores as a novel route for AI delivery in insects that would enhance mortality.

Carbon dots (CDs) have gained significant attention in delivery applications due to their unique properties of facile synthesis, small size, high aqueous dispersibility, internal fluorescence properties, bio-compatibility, and degradability[37,38]. Several studies have demonstrated the beneficial effects of CDs on various plant physiological processes, including growth, photosynthesis, and resistance to biotic/abiotic stress[39–42]. The surface functional groups present on CDs make them easily modifiable, enabling CDs as excellent nanocarriers for the targeted delivery of chemical cargoes[43,44]. In this context, cyclodextrins, a class of non-toxic cyclic molecules made up of glucose units linked by α1-4 glycosidic bonds, is a versatile tool for constructing smart nanodelivery systems in combination with other nanomaterials[45]. These molecular baskets with a hydrophobic internal cavity form an inclusion complex with hydrophobic compounds improving solubilization, slow-release, reduced AI evaporation, and stability in formulations of a wide range of chemicals[46]. Therefore, by incorporating insecticide AIs into cyclodextrin-modified CDs, it may be possible to improve their delivery and increase their efficacy against stink bugs. Tagging chloroplast tar-geting peptides or sucrose molecules to β-cyclodextrin functionalized CDs has been shown to enhance nanocarrier-mediated delivery of fluorescent chemical cargoes into plant organelles (chloroplasts) and vasculature (phloem)[44,47]. CDs nanocarriers with molecular baskets have been studied for chemical cargo delivery in plants but not in insect pests.

In this study, we hypothesized that γ-cyclodextrin-modified car-bon dots (γ-CDs) act as a nanocarrier to deliver insecticide AI into the tarsi of stink bugs (*N. viridula*), thereby increasing insect mortality. The physicochemical properties of the nanocarriers and nanoformulations were designed to restrict their uptake into the leaf, allowing them to remain on the leaf surface. This facilitates delivery to the insects through submicron sized dermal pores and nano-sized cuticular canals (either wax canals or pore canals) present on their tarsi as they walk on the leaf surface (Fig. 1). We demonstrated proof of concept of tarsal delivery of nanocarriers with γ-cyclodextrin functionalized Gd$^{3+}$-doped CDs (γ-GdCDs). The uptake of the fluorescent nanocarriers with chemical cargoes through stink bug tarsi and their ability to remain on the plant leaf surface was elucidated by confocal microscopy and elemental analysis. For agricultural applications, an undoped γ-CDs nanocarrier with comparable physiochemical properties to γ-GdCDs was loaded with insecticide AI (γ-CDs-AI) to evaluate whether this nanoformulation increases stink bug mortality. Nanotechnology-mediated insecticide AI delivery approaches can enable efficient AI delivery, improved integrated pest management, and a more sustain-able agriculture.

## Results
### Synthesis and characterization of nanocarriers
We synthesized and functionalized GdCDs and CDs with γ-cyclodextrin to generate nanocarriers for improving the delivery efficiency of insecticide AI to stink bugs (Supplementary Fig. 1 and Fig. 2a). GdCDs and CDs were coated with boronic acids via the formation of an amide bond between amine groups of CDs and carboxyl groups of 4-carboxyphenyl boronic acid using EDC/NHS coupling[43,44]. The boronic acid functionalized GdCDs and CDs were tethered covalently to the γ-cyclodextrin through the formation of cyclic boronic ester bond between the boronic acid and cis-diols of γ-cyclodextrin[43,48], resulting in γ-GdCDs and γ-CDs, respectively. The CD nanocarrier synthesis and purification methods were selected to minimize, to the extent possible, the number of steps and reagents needed thereby reducing material and labor costs. We estimated the cost for our insecticide nanoformulation, considering various components including precursors, chemical reagents, and purification materials, using current market prices and the amount of material needed per hectare or per plant (SI Table 1). Significant cost reductions of perhaps an order of magnitude could be feasible by purchasing the chemicals in bulk and large-scale synthesis.

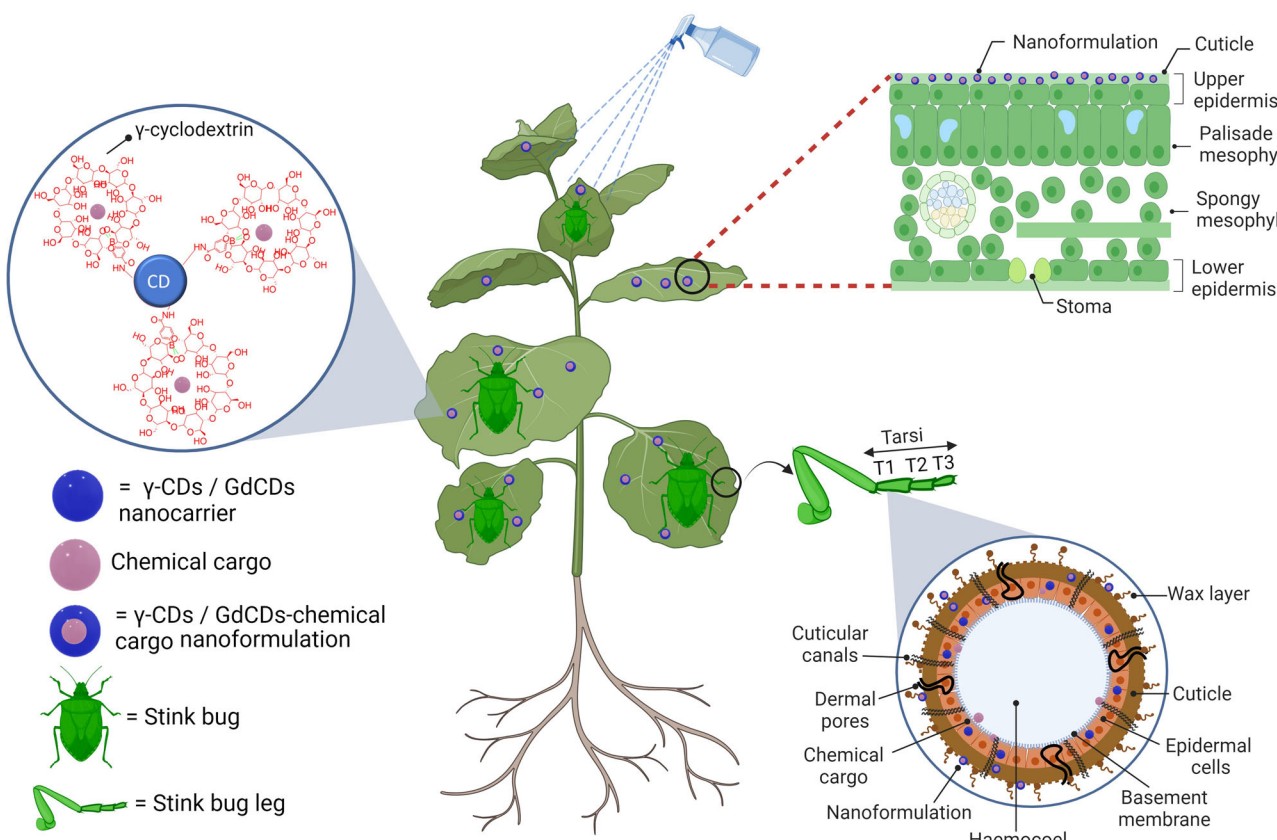

**Fig. 1 | Targeted carbon dot-based nanocarriers for the delivery of chemical cargoes to stink bugs (*Nezara viridula*).** Nanocarriers made of carbon dots (CDs) with molecular baskets (γ-cyclodextrins), were specifically designed with size and charge properties that facilitate their spontaneous translocation through the stink bug leg tarsal surface and effectively reduce the delivery of chemical cargoes through the plant leaf surface. We used Gd[3+] doping of CDs (GdCDs) to detect nanocarriers by elemental analysis in stink bugs. The fluorescence properties of the CDs and their chemical cargoes allow high-resolution imaging by confocal microscopy. This nanotechnology-based approach offers the potential to enhance the delivery efficiency of insecticide active ingredients, while simultaneously reducing losses in the environment. Created in BioRender. Lab, G. (2023) B.

The morphology and size distribution of CDs and GdCDs before and after modification with γ-cyclodextrin were characterized using TEM (Supplementary Fig. 2). Both CDs and GdCDs exhibited spherical shapes, and the size distribution of CDs and GdCDs after modification with γ-cyclodextrin was increased from $5.6 \pm 1$ to $9.0 \pm 1.3$ nm and $6.1 \pm 1.3$ to $9.3 \pm 1.5$ nm ($P < 0.0001$), respectively, which could be due to presence of an organic layer. Similar increases in the size of CDs and metal nanoparticles (NPs) after functionalization with cyclodextrins have been reported previously[43,49]. The thickness of GdCDs and γ-GdCDs were analyzed using AFM (Fig. 2b, c). After functionalization with γ-cyclodextrin, γ-GdCDs showed an increase in thickness from $3.4 \pm 0.2$ nm to $5.8 \pm 0.25$ nm ($P < 0.0001$) (Fig. 2c). This finding is consistent with a previous report of an increase in the lateral height of carbon quantum dots following cyclodextrin modification[50]. Dynamic light scattering analysis revealed that the hydrodynamic size of GdCDs ($5.7 \pm 0.5$ nm) and CDs ($4.7 \pm 0.4$ nm) significantly increased after γ-cyclodextrin functionalization to $8.0 \pm 0.8$ nm ($P < 0.05$) and $7.6 \pm 0.6$ nm ($P < 0.005$) for γ-GdCDs and γ-CDs, respectively (Fig. 2d), which is consistent with the size measured by TEM (Supplementary Fig. 2). The relatively high zeta-potential of GdCDs and CDs showed a value of $-23.6 \pm 1.1$ mV and $-36.8 \pm 1.6$ mV (TES buffer, pH 7), respectively, (Fig. 2e) and can be attributed to the presence of abundant carboxyl groups on their surface[51,52]. However, after functionalization with γ-cyclodextrin, the zeta-potential of γ-GdCDs and γ-CDs decreased in magnitude to $-15 \pm 1.4$ mV ($P < 0.001$) and $-12.6 \pm 0.7$ mV ($P < 0.0001$), respectively, similar to previous reports for carbon dot nanocarriers[44]. The measured values are in close proximity to the reported zeta-potential of cyclodextrin alone[49],

suggesting that γ-cyclodextrin effectively masked the negatively charged groups on the surface of GdCDs and CDs[53,54]. Both the hydrodynamic size and zeta-potential measurements indicated nearly identical physicochemical characteristics of γ-GdCDs and γ-CDs. The presence of cyclodextrin derivatives was also confirmed by Fourier Transform Infrared Spectroscopy (FTIR) analysis (Fig. 2f). FTIR analysis of GdCDs and CDs indicated signature bands at 1180 cm⁻¹, 1355 cm⁻¹, 1635 cm⁻¹ and a broad peak between 3200–3600 cm⁻¹ corresponding to the C-N stretching vibration, O-H bending vibrations, C = O stretching vibrations, and O-H and N-H stretching vibrations, respectively[51,55,56]. Following modifications with 4-carboxyphenyl boronic acid (CPBA), the FTIR spectra of γ-GdCDs and γ-CDs showed additional peaks at 1584 cm⁻¹ corresponding to the amide bond formed between the carboxyl groups of CPBA and amine groups of CDs. The glycosidic vibration bands (C-O-C) at 1042 cm⁻¹ confirmed γ-GdCDs and γ-CDs functionalization with γ-cyclodextrins[21]. Furthermore, ¹¹B NMR spectroscopy was conducted to detect the boron signal from boronic ester bonds established between γ-cyclodextrin and CDs in γ-CDs (Supplementary Fig. 3). This analysis revealed a prominent peak in the spectra of γ-CDs that was not observed in unmodified γ-cyclodextrin or uncoated CDs, validating the successful functionalization of γ-CDs with γ-cyclodextrins.

The optical properties of nanocarriers were characterized using absorbance and fluorescence spectroscopy. GdCDs and γ-GdCDs showed absorbance maxima at 349 nm and 350 nm, respectively (Fig. 2g), which is attributed to the n-π* transition of C = O bands[55], indicating the presence of carboxyl/carbonyl groups on the surface of GdCDs and γ-GdCDs. The fluorescence spectra of GdCDs and γ-GdCDs

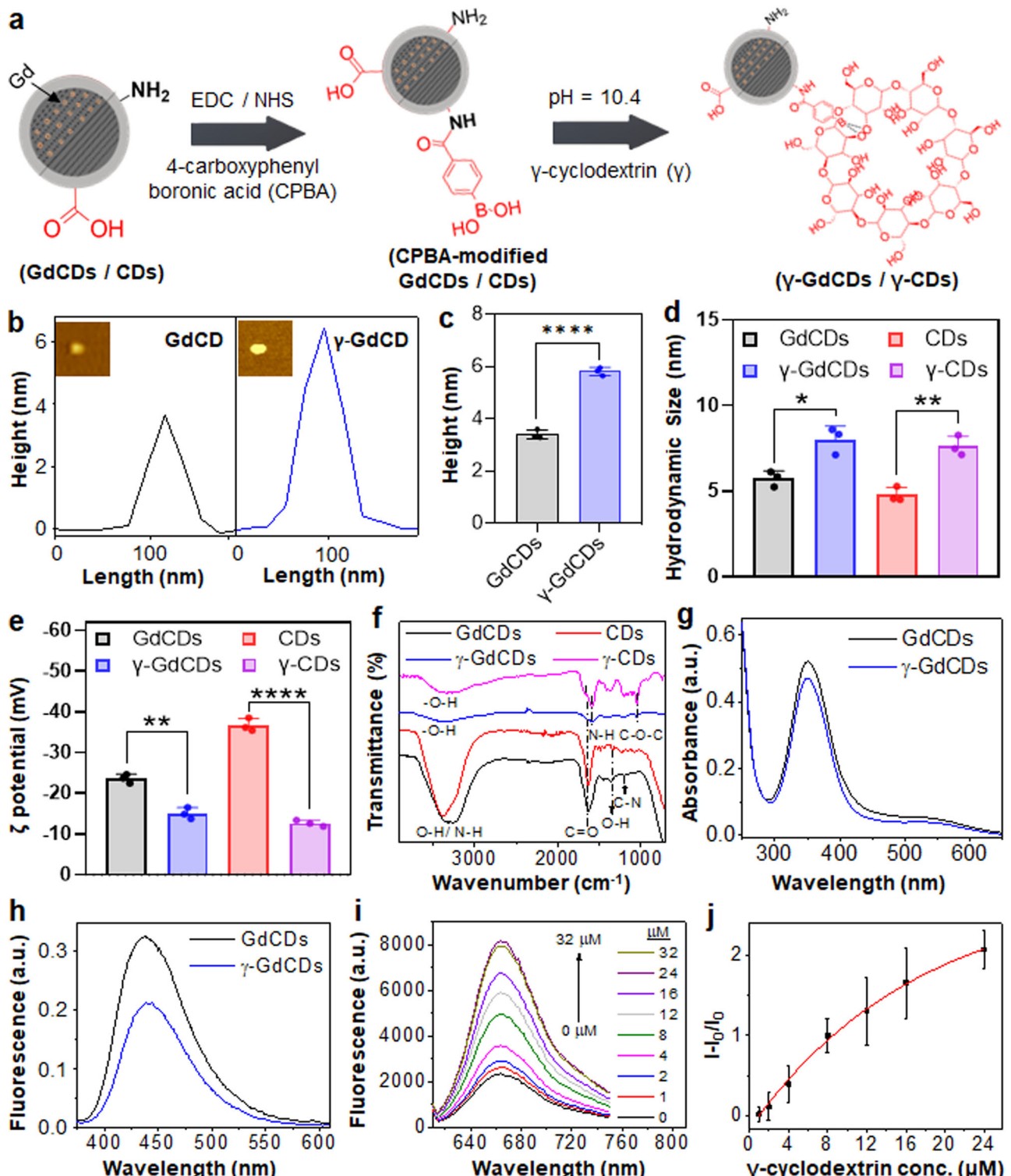

**Fig. 2 | Characterization of nanocarriers made of CDs and molecular baskets and loading of fluorescent cargo. a** Schematic depicts the steps for γ-cyclodextrin modifications of GdCDs (γ-GdCDs) and CDs (γ-CDs). **b** AFM analysis of GdCDs and γ-GdCDs, with (**c**) their average height profile (****$P$ < 0.0001, two-tailed unpaired t-test). **d** Hydrodynamic size (*$P$ = 0.0120, **$P$ = 0.0025, two-tailed unpaired t-test), **e** zeta potential (**$P$ = 0.0011, ****$P$ < 0.0001, two-tailed unpaired t-test), and **f)** FTIR spectra of GdCDs, γ-GdCDs, CDs, and γ-CDs. **g** Absorbance and **h** fluorescence spectra of GdCDs and γ-GdCDs. **i** Fluorescence spectra of Nile red dye (Ex. λ− 585 nm) in the presence of different concentrations of γ-cyclodextrin. **j** Calibration curve showing non-linear fitting between nile red fluorescence intensity changes ($I$-$I_0$/$I_0$) and different concentrations of γ-cyclodextrin. $I$ and $I_0$ represent the fluorescence intensities of Nile red in the presence and absence of γ-cyclodextrin, respectively. Values in panel **c**–**e** and **j** represent means and error bars indicate standard deviation ($n$ = 3 independent replicates).

upon excitation at 355 nm showed the emission maxima at 438 nm and 441 nm, respectively (Fig. 2h). The red shift in the emission maxima of γ-GdCDs can be ascribed to the quantum size effect[57]. Further, γ-cyclodextrin functionalization caused the quenching of GdCDs fluorescence intensity (measured at the same concentration of Gd) by 1.5 times. Similar quenching in the fluorescence intensity of CDs after modification with cyclodextrin and other capping agents has been reported before[58]. In contrast, CDs and γ-CDs showed two absorbance peaks at 334 and 394 nm (Supplementary Fig. 4a), which can be assigned to the n-π* transition of C = O bands and formation of conjugated sp$^2$ carbon cores in CDs, respectively[59]. Furthermore, both CDs and γ-CDs demonstrated fluorescence emission maxima at 450 nm (Supplementary Fig. 4b). Using quinine sulfate (QY = 54%) as a reference, the GdCDs showed a relatively high quantum yield (QY) (59%) as compared to CDs (1.7%), which may be attributed to the disruptive effect of Gd on the carbon rings, leading to the creation of new energy traps for emission[55,60]. This high QY along with the presence of traceable Gd element in GdCDs allowed fundamental research about their interactions with plant leaf surfaces by confocal microscopy and uptake in stink bug tarsi by ICP-OES analysis.

The fraction of γ-cyclodextrin in γ-GdCDs and binding to fluorescent chemical cargoes for fluorescence microscopy studies in insects was determined by investigating the host-guest interaction between Nile red dye and γ-cyclodextrin. To conduct this study, we determined the solubility of Nile red dye in varying DMSO concentrations in aqueous solution through spectrophotometric analysis. With a decrease in DMSO percentages from 100% to 50%, the dye solution showed a red shift in the absorbance maxima (from 550 nm to 582 nm), without a large reduction in absorbance intensity (Supplementary Fig. 5a). However, a significant decrease in the dye absorbance in 30% DMSO was observed, which may be attributed to the aggregation of dye with decreasing DMSO content. Furthermore, dye aggregation was observed through a color change of the solution (from pink-blue-transparent) with decreasing DMSO percentage (Supplementary Fig. 5b). The aggregation of dye is reported to be prevented by forming the inclusion complex with cyclodextrins[61]; therefore a 30% DMSO solution containing fixed amounts of dye and varying amounts of γ-cyclodextrin was used to study the host-guest interactions based on the changes in dye fluorescence intensity. We observed a gradual increase in the emission intensity of dye with increasing γ-cyclodextrin concentration from 0 μM to 24 μM and the final attainment of a plateau at 32 μM (Fig. 2i). This phenomenon is attributed to the increase in dye solubility facilitated by the formation of an inclusion complex with γ-cyclodextrin[62]. Using a calibration curve generated by fitting the nonlinear relationship between changes in dye fluorescence intensity ($I$-$I_0$/$I_0$) and different concentrations of γ-cyclodextrin (Fig. 2j), the fraction of γ-cyclodextrin in γ-GdCDs was calculated to be 11.8 wt.%. Considering a 1:1 host-guest stoichiometry[63], the content of Nile red dye in γ-GdCDs was calculated to be 3 wt.%.

For determining the efficacy of nanocarrier-mediated delivery of AI in agricultural applications, we synthesized undoped nanocarriers without Gd (γ-CDs) and loaded them with an insecticide AI (γ-CDs-AI) (see methods). The formation of inclusion complexes using cyclodextrins can offer several advantages to the guest AI molecules, such as enhanced solubility, stabilization of the molecule in solution, and reduced losses due to volatilization[64]. After loading the AI, the γ-CDs-AI exhibited no notable change in the hydrodynamic size (7.0 ± 0.4 nm) (Supplementary Fig. 6a) and zeta potential (−14.9 ± 1.3 mV) in comparison to γ-CDs (Supplementary Fig. 6b). This can be attributed to the efficient complexation of AI with γ-CDs, without any significant adsorption occurring on the γ-CDs surface[65]. The FTIR spectrum of γ-CDs-AI shows the appearance of several new peaks that match the peaks of AI alone, in addition to the peaks corresponding to γ-CDs (Supplementary Fig. 7). However, a few AI peaks at 2966, 1726, and 1350 cm$^{-1}$ were shifted to 2950, 1716, and 1337 cm$^{-1}$, respectively, in the

γ-CDs-AI spectrum. Additionally, in comparison to the γ-CDs spectrum, we observed the glycosidic vibration (C-O-C) peak of γ-cyclodextrin in the spectrum of γ-CDs-AI had shifted from 1042 cm$^{-1}$ to 1072 cm$^{-1}$. These spectroscopic changes indicate the inclusion complex formation between γ-CDs and AI. The loading capacity of γ-CDs for the AI was quantified as 14.4 ± 1.5%, after liquid-liquid AI extraction and quantification using a UV-visible spectrophotometer. The formation of the inclusion complex between γ-CDs and AI is primarily attributed to hydrophobic interactions between the inner hydrophobic cavity of γ-cyclodextrin and the hydrophobic AI[66]. The loading and delivery of hydrophobic AI in γ-CDs-AI nanoformulations could improve AI mortality efficacy for *N. viridula* due to the enhanced solubility and stabilization of the hydrophobic AI[67,68].

The selection of γ-cyclodextrin over other cyclodextrins (α and β) was based on the larger cavity size of γ-cyclodextrin that allows the loading of the Nile red dye and insecticide AI as described above. Minimal host-guest interactions were observed between the Nile red dye (318.37 g/mol) and the smaller cavity sized β-cyclodextrins (Supplementary Fig. 8). The loading capacity of AI (500 g/mol) in β-cyclodextrin CDs was determined to be only 1.4 ± 0.6%. In addition, γ-cyclodextrins have been reported to be more biocompatible than the other α- and β-cyclodextrins because of their lower impact on cellular lipids[69,70].

### Restricted nanocarrier and chemical cargo translocation across the plant leaf surface

The cuticle and stomatal pores on the leaf epidermal layer act as the main pathways for the uptake of NPs into plant leaves[71]. We designed nanocarriers and nanoformulations that restrict uptake into the leaf to increase the availability of AI to the stink bugs. Various physicochemical properties of NPs, such as size, charge, hydrophobicity, and aspect ratio affect their uptake and translocation in plants[20,26,71,72]. The surfactant surface tension in nanoformulations also affects the uptake of NPs across the leaf surface[20]. To investigate the translocation of nanocarriers and their chemical cargoes on the plant leaf surface, γ-GdCDs were loaded with R6G dye (γ-GdCDs-R6G) (Supplementary Fig. 9). Soybean leaves treated with γ-GdCDs-R6G (suspended in 0.1% triton x-100), displayed fluorescence from both γ-GdCDs (cyan) and R6G dye (yellow) on the epidermal surface but not from inside leaf mesophyll tissue containing chloroplasts (Fig. 3a). A Pearson's correlation (PCC) analysis[73–75], confirmed no correlation (PCC = 0) between γ-GdCDs and R6G dye fluorescence with the chloroplast autofluorescence. Instead, a high colocalization of γ-GdCDs with R6G dye fluorescence (PCC = 0.57 ± 0.12) indicated that nanocarriers effectively retain the fluorescent cargo and prevent its release onto the leaf surface. Control leaves treated solely with 0.1% triton x-100 displayed no background autofluorescence in the γ-GdCDs nanocarrier and R6G emission channels (Fig. 3a). Similarly, orthogonal views of z-stack images (Fig. 3b) of leaves treated with γ-GdCDs-R6G showed no overlap of γ-GdCDs and R6G dye fluorescence with chloroplast autofluorescence, but instead the colocalization of γ-GdCDs with R6G dye fluorescence. Reconstructed 3D images from z-stacks confirmed the localization of nanocarrier and R6G dye fluorescence over the leaf surface (Fig. 3c). In contrast, leaves treated with chemical cargo (R6G) alone (with 0.1% triton x-100), showed colocalization of R6G fluorescence with chloroplast autofluorescence inside the leaf mesophyll (Supplementary Fig. 10). Overall, confocal microscopy analysis indicates that γ-GdCDs nanocarriers with their chemical cargo (R6G) preferentially localize on the leaf epidermal surface.

Translocation analysis of γ-GdCDs from treated leaves to different parts of soybean plants based on ICP-OES analysis indicated that most nanocarriers remain on the leaf surface after two days of exposure. After 24 h of exposure, the amount of γ-GdCDs inside plants was 11.7 ± 1.6%, with 2.4 ± 0.8% in treated leaves, 4.0 ± 1.6% in untreated leaves, 4.9 ± 1.3% in shoots, and 0.4 ± 0.04% in roots (Fig. 3d). After

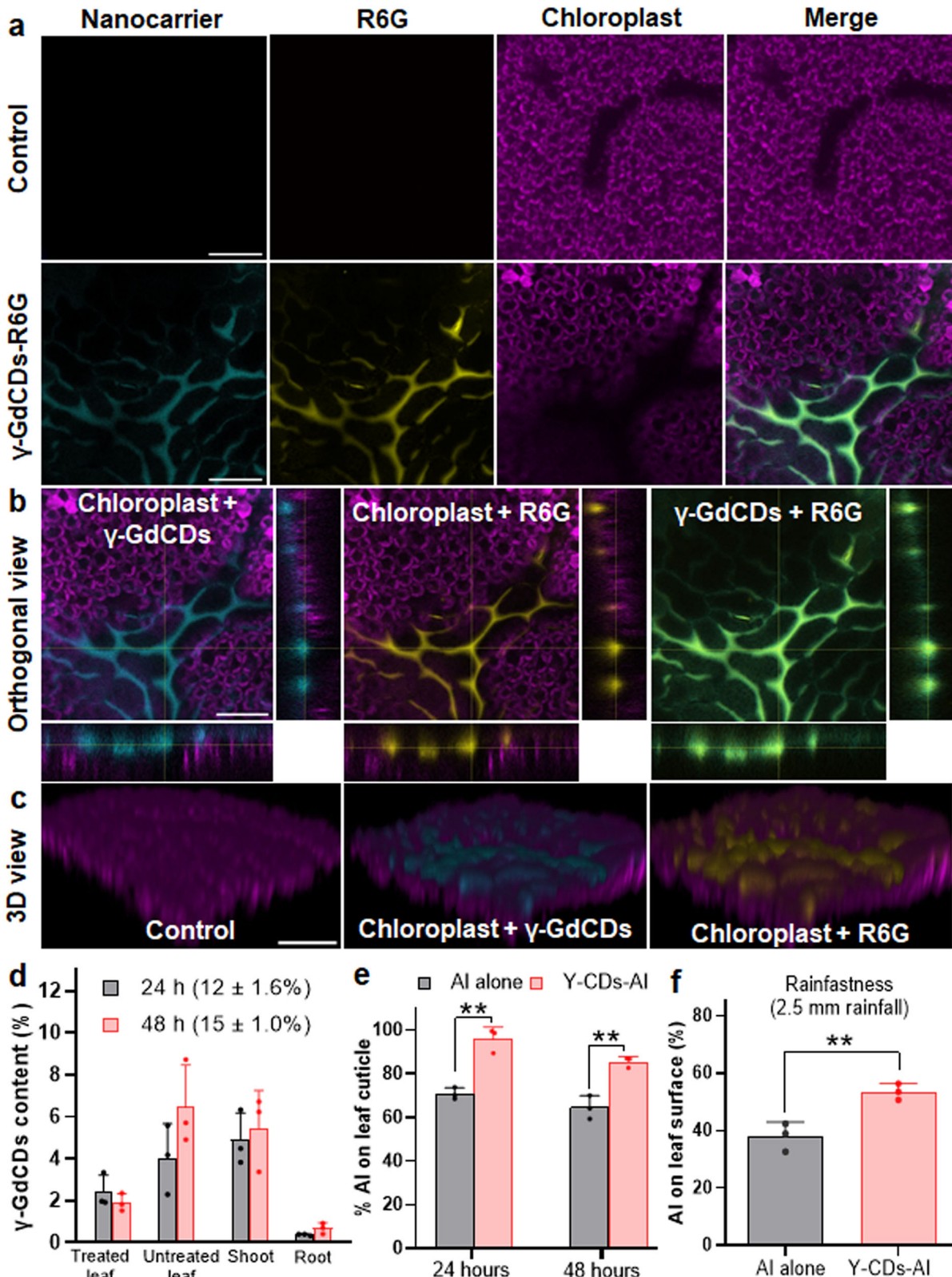

**Fig. 3 | Nanocarrier and chemical cargo interactions with plant leaf surface.**
**a** Confocal images of leaves after 3 h incubation with nanocarriers. Cyan indicates γ-GdCDs fluorescence (Ex. λ− 355 nm), yellow indicates R6G dye fluorescence (Ex. λ− 488 nm), and magenta indicates chloroplast autofluorescence. Green indicates the colocalization of γ-GdCDs with R6G dye. Scale bar = 50 μm. **b** Orthogonal and **c** 3D views of representative confocal z-stacks displaying no colocalization of nanocarrier and dye with chloroplasts in leaf mesophyll cells but instead the colocalization of γ-GdCDs with R6G dye (n = 3). Scale bar = 100 μm. The experiments were independently repeated three times with similar results. **d** ICP-OES analysis of γ-GdCDs content in different plant parts. **e** LC-MS analysis of AI on leaf cuticular surface treated with γ-CDs-AI and AI alone (**P = 0.0022 (24 h), **P = 0.0035 (48 h), two-tailed unpaired t-test). **f** Leaf surface rainfastness analysis of γ-CDs-AI and AI alone at 2.5 mm rainfall (**P = 0.0092, two-tailed unpaired t-test). Values in panel **d**–**f** represent means and error bars indicate standard deviation (n = 3 biological independent replicates).

48 h, the amount of γ-GdCDs inside plants was $14.5 \pm 1.0\%$, similar to 24 h exposure ($P > 0.05$), with $1.9 \pm 0.4\%$ in treated leaves, $6.5 \pm 2.0\%$ in untreated leaves, $5.5 \pm 1.8\%$ in shoots, $0.6 \pm 0.26\%$ in roots. After two days of application, over 85% of the nanocarriers remained on the leaf surface. Similarly, most of AI remained on the soybean leaf cuticular surface after being treated either with the AI loaded γ-CDs (γ-CDs-AI) or AI alone, as quantified by LC-MS (Fig. 3e). The cuticular wax on the leaf surface acts as a permeation barrier for the uptake of AI[76]. After 24 h of exposure, γ-CDs-AI exhibited notably higher AI levels ($95.8 \pm 5.5\%$) on the leaf cuticular surface compared to AI alone ($70.9 \pm 2.7\%$) ($P < 0.05$). After 48 h, the AI percentage for both γ-CDs-AI and AI alone treatments decreased to $85.5 \pm 2.4\%$ and $64.6 \pm 5.3\%$ ($P < 0.05$), respectively. The decrease in AI content by 10% on the leaf surface for the γ-CDs-AI, may be attributed to the release of AI from γ-CDs-AI followed by AI penetration inside the leaf. Nonetheless, the significantly higher AI percentage retained by γ-CDs-AI after two days confirmed the nanocarrier's superior AI retention ability on the leaf surface compared to AI alone.

The nanocarrier localization on the leaf surface could be attributed to cuticle size exclusion limit, surfactant surface tension, or repulsive electrostatic interactions with the cell wall. The leaf cuticle possesses <2 nm hydrophilic pores[77–79], which may impose a size exclusion limit and prevent the uptake of ~8 nm size γ-GdCDs nanocarriers. In addition, the high surface tension of triton x-100 could prevent the uptake of γ-GdCDs through stomatal pores into the leaf mesophyll. A CD dispersion in triton x-100 surfactant, with a high surface tension of 30 mN/m, prevented CDs uptake in both dicot and monocot plants; whereas CDs dispersion in Silwet L-77 surfactant, with a low surface tension of 22 mN/m, allowed CD uptake[20]. Repulsion electrostatic interactions between the negatively charged cell walls and negatively charged γ-GdCDs may also inhibit nanocarrier translocation across the leaf epidermis. Pectin in plant cell walls has a negative charge[80], exhibiting a higher affinity towards positively charged NPs[81]. Furthermore, based on NP-leaf interaction empirical models, NPs with a charge below +15 mV exhibited less foliar uptake efficiencies into mesophyll tissue[20]. This nanocarrier and nanoformulation design carrying loaded AI prevents their uptake into soybean leaves making them readily available to stink bugs on the leaf surface.

Rainfall contributes to the loss of insecticide AI in agricultural fields[82]. Therefore, we investigated the rainfastness efficiency of γ-CDs-AI in comparison to AI alone on soybean leaves under simulated rainfall conditions of 2.5 mm and 5 mm. The γ-CDs-AI demonstrated significantly higher retention of AI on the leaf surface ($54 \pm 3\%$) compared to AI alone ($38 \pm 5\%$) ($P < 0.05$) under 2.5 mm rainfall (Fig. 3f). However, in our simulated 5.5 mm rainfall experiment, over 90% of the AI was lost for both the nanocarrier and AI alone treatments (Supplementary Fig. 11). The moderate retention of AI by γ-CDs-AI during light rainfall events may be due to the weak interactions of γ-CDs with the leaf cuticle containing polysaccharides and waxes. The n-π* transition of the C = O bond at 334 nm in CDs (see Supplementary Fig. 4a) can form π-π stacking interactions with the leaf cuticle wax aromatic compounds (phenylpropanoids and polyphenols)[83,84]. Also, -COOH groups of CDs, and -OH groups of γ-cyclodextrin may interact via H-bonding formation with the exposed -OH groups of leaf cuticle wax aliphatic compounds and cuticle polysaccharides[84]. Although γ-CDs-AI can mitigate AI loss under light rainfall conditions, they may not be as effective under moderate to heavy rainfall.

### Fluorescent chemical cargo delivery to stink bugs tarsi by nanocarriers

To assess the nanocarrier-mediated delivery of a model fluorescent chemical cargo through the stink bug tarsi, we characterized the optical and morphological properties of *N. viridula* tarsi using confocal laser scanning microscopy (CLSM) and scanning electron microscopy

(SEM). Upon excitation at 355 nm and 488 nm wavelengths, the tarsi exhibit bright autofluorescence in both UV-blue (400–500 nm) and green-yellow (500–600 nm) visible spectra regions (Fig. 4a). UV-blue fluorescence indicates exoskeleton structures that predominantly consist of the soft and highly elastic protein known as resilin; whereas green-yellow fluorescence indicates exoskeleton structures composed of weakly or nonsclerotised chitinous material[35]. However, upon excitation at 561 nm wavelength, the tarsi do not exhibit any autofluorescence in the orange-red fluorescence region (600–700 nm). Similarly, 50 μm cryostat sections of tarsi show bright autofluorescence at 355 nm and 488 nm excitation wavelengths in UV-blue and green-yellow fluorescence range, but no autofluorescence at 561 nm excitation within the orange-red fluorescence range (Supplementary Fig. 12). This no background fluorescence range offers a fluorescence spectra window upon 561 nm excitation for imaging nanocarrier mediated cargo delivery of a fluorescent chemical cargo into the insect tarsi.

To determine the morphological features of the tarsal surface, we characterized the surface of *N. viridula* tarsi using SEM (Fig. 4b). The tarsi of *N. viridula* are composed of three tarsal segments: basitarsus (T1), mediotarsus (T2), and distitarsus (T3) (Fig. 4b-i, ii). The ventral surface of the pulvilli appears smooth (Fig. 4b-iii), but upon closer examination (Fig. 4b-iv), it becomes apparent that it contains grooves running parallel to the longitudinal axis of the pretarsus. Other studies have identified numerous pores on the ventral surface of the pulvillus[35]. The ventral surface of T1 at low magnification (2500 x) showed what appears to be micron-size openings of $4.5 \pm 1.6 \mu m$ in length (Fig. 4b-v). At 4000–8000 x, the cuticle shows what seems to be sub-micron size dermal pores ranging from $0.38 \pm 0.1 \mu m$ (Fig. 4b-vi) to $0.16 \pm 0.02 \mu m$ (Fig. 4b-vii). At 5000× magnification, there is a wide distribution of putative pores and canals with an average length of $770 \pm 280$ nm and a width of $160 \pm 37$ nm (Fig. 4b-viii). At even higher magnifications (80,000×), nano-sized cuticular canals of $55 \pm 12$ nm in length were observed to be distributed more uniformly (Fig. 4b-ix) at a density of $43.55 \pm 9.20/\mu m^2$ [35]. These tarsi structures can come in contact with the leaf surface[85,86] and the putative pores and canals ranging from micron to nanoscale size may act as uptake pathways for CDs nanocarriers and AI through the tarsi.

To determine nanocarrier mediated delivery of chemical cargoes into stink bug tarsi, we loaded γ-GdCDs nanocarriers with Nile Red fluorescent dye (γ-GdCDs-Nile red) and performed confocal microscopy analysis of the tarsal sections (Supplementary Fig. 13). Nile Red was chosen for its non-polar nature like many insecticide AIs, and it possesses excitation and emission maxima at 561 nm and 635 nm, respectively, which allowed imaging in the insect tarsi without background autofluorescence (see Fig. 4a). Insect tarsi treated solely with 0.1% triton x-100 display bright autofluorescence (UV-blue range) when excited at 488 nm, whereas no background autofluorescence was observed in the Nile red emission channels (Fig. 5a). However, the confocal images of T1 sections from proximal and distal ends treated with Nile red and γ-GdCDs-Nile red demonstrated the uptake of both Nile red dye alone and delivered by γ-GdCDs (Fig. 5a). We selected T1 for the uptake analysis because this segment along with distal portions of the pulvilli are in direct contact with the leaf surface during stink bug walking[85,86], and the highly curved pulvilli impede preparation of sections for imaging. Tarsal sections prepared from both proximal and distal ends after treatment with Nile red (Supplementary Movie 1, 2) and γ-GdCDs-Nile red (Supplementary Movie 3, 4), confirmed colocalization of Nile red with the tarsal autofluorescence throughout the 50 μm z-stack. Insects treated with γ-GdCDs-Nile red nanocarriers exhibit 2.6-fold higher fluorescence intensity of Nile red in both the proximal and distal ends of the tarsi than insects treated solely with Nile red ($P < 0.05$) (Fig. 5b), demonstrating the capacity of nanocarriers to enhance the delivery of a chemical cargo. The higher

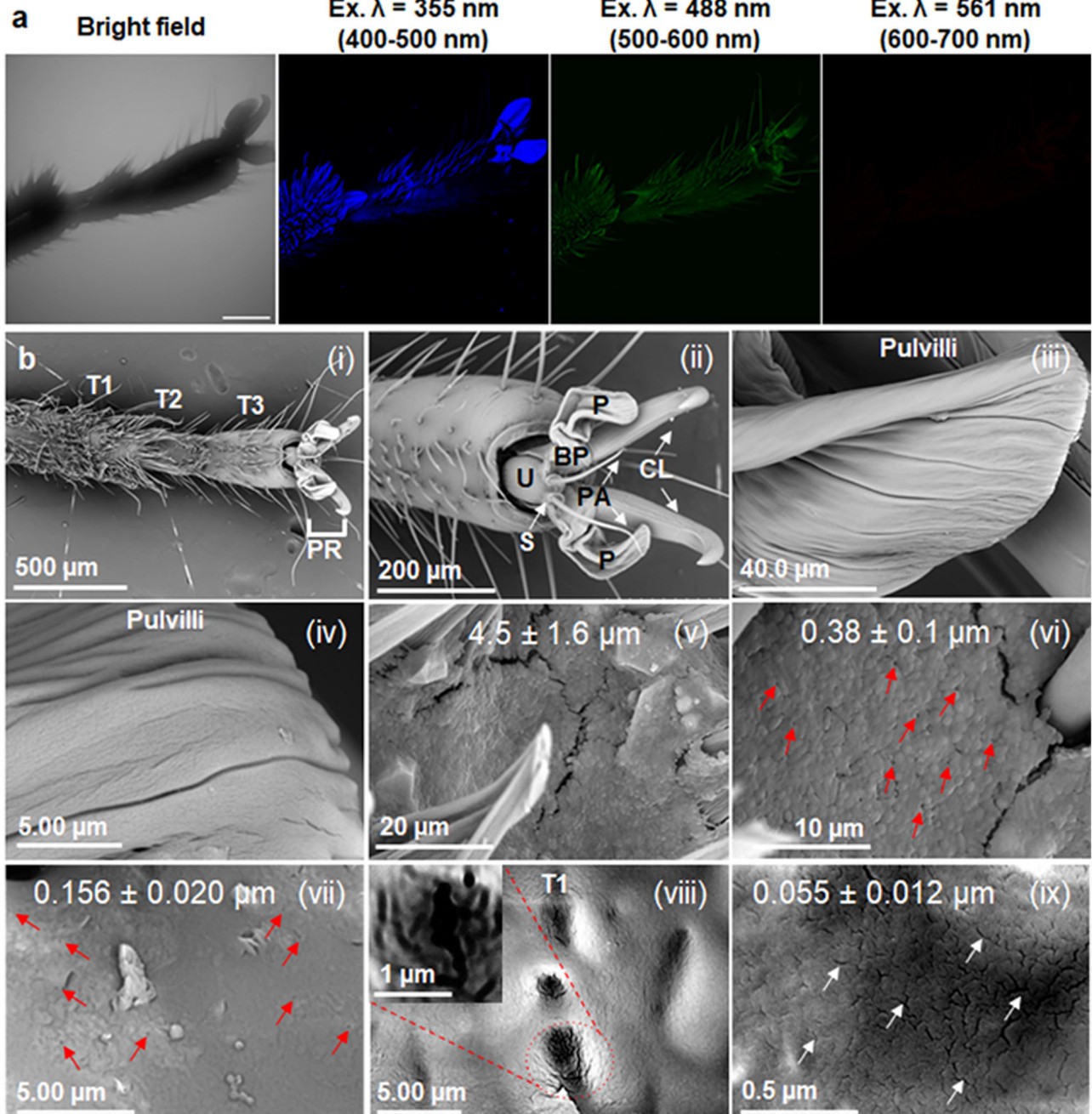

**Fig. 4 | Optical and physical properties of stink bug tarsi. a** Confocal images of *N. viridula* tarsi show autofluorescence after excitation at 355 nm and 488 nm wavelengths but not at 561 nm excitation wavelength (scale bar = 250 μm). **b** SEM images of *N. viridula* tarsi. (i) Ventral view of three tarsal segments (T1-T3) and pretarsus (PR). (ii) Ventral view of PR showing an unguitractor plate (U), basipulvillus (BP), two pulvilli (P), two curved claws (CL), and two parapodia (PA). (iii-ix) High magnification images of pulvilli (iii,iv) and T1 (v-ix). Arrows indicate putative dermal pores (red) and cuticular canals (white) present on the tarsal surface. Inset in (viii) shows the magnified image of cuticular canals highlighted in red circles. The experiments were independently repeated three times with similar results.

uptake in the γ-GdCDs-Nile red nanocarriers could be attributed to the presence of the pores and cuticular canals on the tarsal surface (see Fig. 4b-v-ix), which allows uptake of nanosized γ-GdCDs-Nile red. Furthermore, encapsulation of Nile red in γ-GdCDs may increase its solubility and thus enhance bioavailability[87]. The uptake of γ-GdCDs in stink bugs was also determined using ICP-OES based on the measurement of the Gd content. Gd was detected in the tarsi of insects treated with nanocarriers, whereas no Gd signal was observed in untreated insects, providing further evidence of γ-GdCDs uptake (Fig. 5c). Our findings are consistent with a recent study demonstrating that NPs (<10 nm) entered pore canals (6–65 nm) in the

insect cuticle of the stored grain pest beetle (*Tenebrio molitor*) and accumulated in the hemocytes[28]. This demonstrates the potential of nanocarriers to overcome insect AI delivery barriers through the stink bug tarsi.

We assessed the uptake of γ-GdCDs via tarsi cuticular canals by detecting their Gd signal using a scanning electron microscope equipped with energy-dispersive X-ray analysis (SEM-EDX). The stink bugs were held overnight in a Petri dish that had been sprayed with nanocarriers (2 mg mL⁻¹). Then, they were transferred to soybean leaves placed with the abaxial side down on 1% agar plates for 2 days. The tarsi were analyzed for the presence of Gd on the cuticular canal

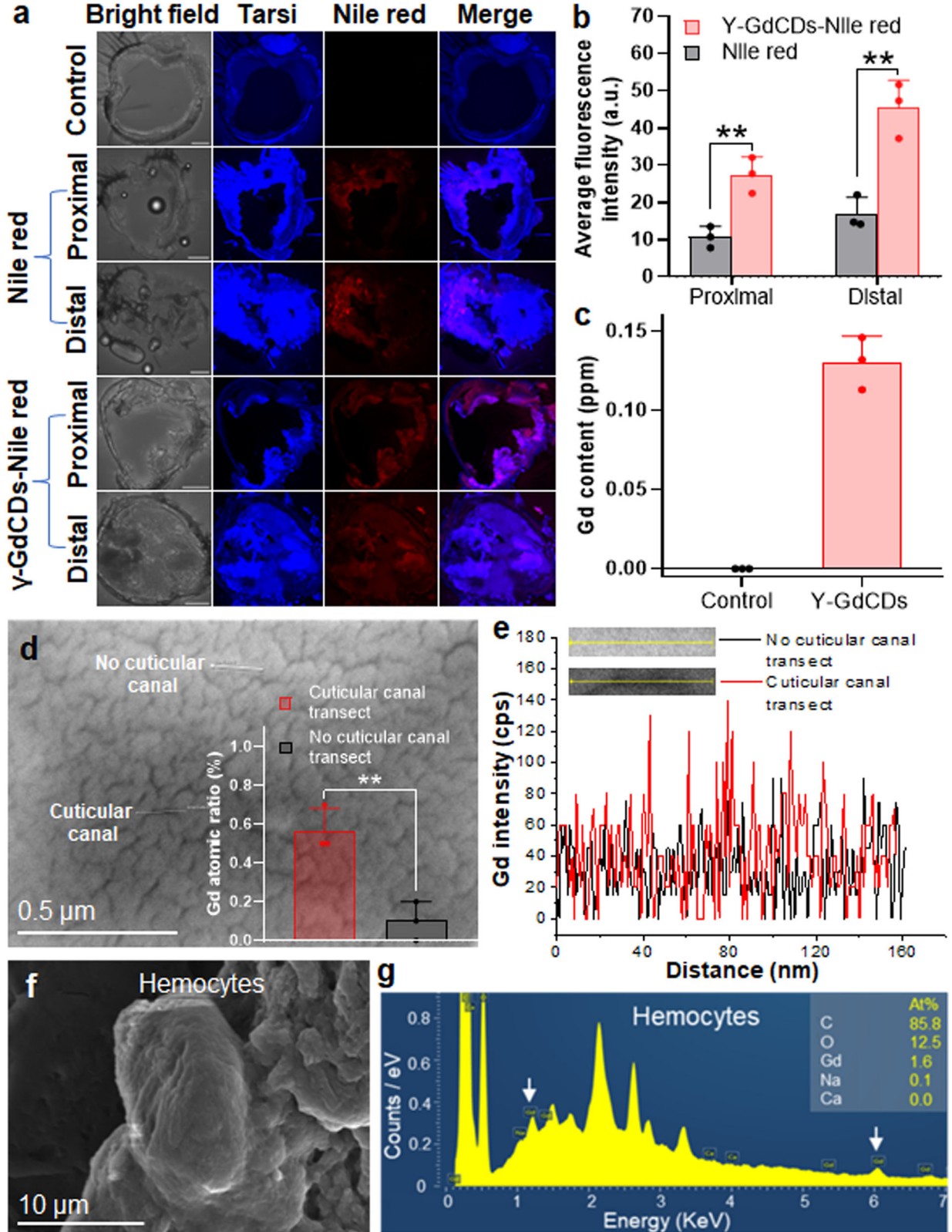

and no cuticular canal transects using an EDX line scan (Fig. 5d, e). We observed a higher Gd signal on the cuticular canal transects in comparison to no cuticular canal counterparts (Fig. 5e). The EDX spectrum indicated a Gd peak (Supplementary Fig. 14a) and significantly higher percent of Gd atomic ratio ($0.57 \pm 0.12$) in the cuticular canal transect compared to no cuticular canal ($0.1 \pm 0.1$) ($P < 0.05$) (Supplementary Fig. 14b and inset in Fig. 5d). To confirm

the uptake of nanocarriers through the tarsi cuticle, we also extracted hemocytes from the hemolymph collected after pricking the insect's forelegs and measured the Gd signal using SEM-EDX (Fig. 5f, g). EDX spectra revealed the presence of Gd signal in hemocytes, confirming the uptake of nanocarriers via the tarsi (Fig. 5g). No Gd signal was detected on the hemocytes of untreated insect tarsi (Supplementary Fig. 15a, b).

**Fig. 5 | Nanocarrier uptake and fluorescent chemical cargo delivery to stink bugs tarsi. a** Confocal images of tarsi sections (T1) prepared from proximal and distal ends show the uptake of nile red dye delivered by γ-GdCDs nanocarriers (*n* = 3). Blue indicates the tarsi autofluorescence (ex. λ = 488 nm) and red indicates Nile red dye fluorescence (ex. λ = 561 nm) (scale bar = 50 μm). **b** Average fluorescence intensity of Nile red delivered without and with γ-GdCDs nanocarriers (**P* = 0.0067 (proximal), **P* = 0.0047 (distal), two-tailed unpaired t-test). **c** Uptake of nanocarriers by the stinkbug tarsi was determined by the ICP-OES analysis of Gd from the γ-GdCDs core. **d, e** EDX line scan performed on SEM images indicate higher Gd signal on the cuticular canal transects than on the no cuticular canal transects of the insect tarsi surface. Inset in (**d**) shows significantly higher percent of Gd atomic ratio in cuticular canal transect than no cuticular canal transects (**P* = 0.0061, two-tailed unpaired t-test). **f, g** SEM-EDX performed on the hemocytes extracted from hemolymph after pricking insect's forelegs indicate the presence of Gd signal (white arrows) and confirms the uptake of γ-GdCDs via tarsi (*n* = 3). Values in panel **b–d** represent means and error bars indicate standard deviation (*n* = 3 biological independent replicates).

## Enhanced mortality of insecticide AI delivered by γ-CDs nanocarriers

The efficacy of the γ-CDs-AI nanoformulation compared to AI alone was tested by exposing insects on soybean leaves sprayed with γ-CDs alone, AI alone, γ-CDs-AI (all mixed with 0.1% Triton x-100), and 0.1% Triton x-100 alone (Supplementary Fig. 16). These treatments were performed at an optimized AI concentration of 10 ppm. 48 h after treatment, the γ-CDs-AI treatment had significantly higher mortality (23.3 ± 5.7%) than the AI alone treatment (13.4 ± 5.8%) (*P* < 0.001) (Fig. 6a). At 72 h, the γ-CDs-AI and AI alone resulted in 63.4 ± 11.5% and 36.7 ± 5.7% mortality, respectively (*P* < 0.0001). By 96 h, the γ-CDs-AI and AI alone caused 83.4 ± 5.8% and 60 ± 10% mortality, respectively (*P* < 0.0001), demonstrating a 25% greater mortality using γ-CDs-AI. A Kaplan-Meier survival analysis test confirmed that γ-CDs-AI caused significantly higher mortality of stink bugs compared to AI alone (Supplementary Fig. 17). Little to no mortality was observed in the Triton x-100 treatment and the γ-CDs alone treatment (Fig. 6a). Overall, these results indicate the superior efficacy of the developed nanoformulation for *N. viridula* mortality compared to AI alone.

The higher mortality with γ-CDs-AI could be attributed to the effectiveness of γ-CDs-AI entering the stink bugs through their tarsal cuticular canals, delivering higher levels of AI compared to AI alone. To test this hypothesis, we evaluated the efficacy of our treatments on insects after removing their stylets, the primary route by which current insecticides are delivered (Supplementary Fig. 18). Stylet removal caused slightly higher, but not statistically different, mortalities in the Triton x-100 and γ-CDs alone treatments after 72 h (Fig. 6b). In contrast, in the AI alone treatment, stylet removal reduced mortality compared to the intact stylets experiment. 48 h after treatment, the AI alone treatment had just 3.4 ± 5.7% mortality, while the γ-CDs-AI had 23.4 ± 5.7% mortality in stink bugs without stylets (*P* < 0.0001). After 72 h and 96 h, the AI alone treatment had 16.6 ± 5.7% and 43.3 ± 5.7% mortality, while the γ-CDs-AI treatment had a significantly higher mortality, 50 ± 10%, and 90 ± 10%, respectively (*P* < 0.0001) (Fig. 6b). A follow-up analysis compared the efficacy of γ-CDs-AI and AI alone on insects with and without stylets. Throughout the 96 h observation period, the γ-CDs-AI had nearly identical mortalities in insects with or without stylets (Fig. 6c), strongly suggesting uptake of γ-CDs-AI into the tarsi. In contrast, in the AI alone treatment there was significantly lower mortality at 48 h (*P* < 0.01), 72 h (*P* < 0.0001), and 96 h (*P* < 0.001) in the insects that had their stylets removed compared to those with stylets (Fig. 6d). This reduced mortality is attributed to the unavailability of AI through the stylet (due to its removal) and the limited AI movement via the tarsi. These findings strongly support the hypothesis that the γ-CDs-AI nanoformulation delivers AI through the insect tarsi leading to superior efficacy in *N. viridula* mortality compared to AI alone. Our study indicates that an alternative and innovative insecticide delivery method through the tarsi, rather than the classical route of the stylets, is successful in achieving high insect mortality.

## Discussion
We developed carbon dot-based nanocarriers (γ-CDs-AI) for the enhanced delivery and efficacy of insecticide AI to the stink bug

*N. viridula* through the tarsi. The physicochemical properties of the nanocarrier maximize the nanoformulation levels on the soybean leaf surface, allowing it to enter the stink bugs via sub-micron size pores in their tarsi as they walk on the leaf surface. Elemental and confocal microscopy analysis indicated that γ-GdCDs are uptaken through the insect tarsi and the delivery of a fluorescent chemical cargo (Nile red dye) is enhanced 2.6 times by the nanocarriers. The γ-CDs-AI increased the solubility of the hydrophobic AI cargo, making it a more efficient formulation for crop protection against pests. Nanocarrier mediated delivery of AI resulted in 25% higher mortality in stink bugs than the AI alone. Styletectomy studies indicated AI delivery through the tarsi by γ-CDs-AI had ~45% higher mortality compared to AI alone in insects with stylets removed. The γ-CDs-AI had a similar mortality regardless of stylet removal, while AI alone showed 20% lower mortality in insects with their stylet removed. This study demonstrates that carbon dot-based nanocarriers are a promising approach for significantly enhancing the delivery and efficacy of insecticide AI through a novel non-classical route.

The utilization of γ-CDs-AI has the potential to revolutionize the delivery of AI directly to insect pests on the leaf surface, resulting in a more accurate and efficient pest management approach. Improving the delivery of pesticide AI in agriculture in a sustainable way will require nanomaterials that are scalable, economical, and have a low environmental footprint[88]. Herein, we used carbon dot nanocarriers, one of the most biocompatible nanomaterials that can be manufactured in a large scale through simple bottom-up approaches[38], using abundant, low-cost, and renewable resources such as animal and plant derivatives[89]. Future research involving the impact of γ-CDs-AI on various immature life stages could expand on the benefits and reduce the costs of nanocarrier-mediated delivery of insecticide AI technologies. Field trials are also needed to identify and mitigate risks pertaining to human health and environmental safety, and to comply with regulations that vary depending on the country of use. This insecticide nanocarrier advancement can lead to substantial reductions in the overall amount of AI required to effectively manage pests like *N. viridula* and other insects with tarsal pores. The "Green Deal" approved by the European Commission aims to reduce the use and risk of chemical pesticides by 50% by 2030[90]. Thus, a 25% increase in insecticide efficiency is highly significant given the potential reductions in the use and risk of insecticides, and in costs related to insecticide applications in the field. Further research is also necessary to examine the effects of γ-CDs-AI on non-target insects. The impact of insecticide AI delivery on pollinators such as bees should be addressed. Adult bees lack dermal glandular pores present in stink bugs[91], presenting an opportunity to design nanocarriers that selectively target harmful insects while having minimal effects on beneficial pollinators. Understanding all these impacts will determine the extent to which this nanotechnology can be implemented in integrated pest management systems for a more sustainable agriculture.

## Methods
### Characterization of nanocarriers
The morphology of CDs and GdCDs was characterized using a transmission electron microscope (TEM) (Thermo Scientific, Talos L120C, Velox Interfaces 3.1 software) at 120 KV. The sample for TEM analysis

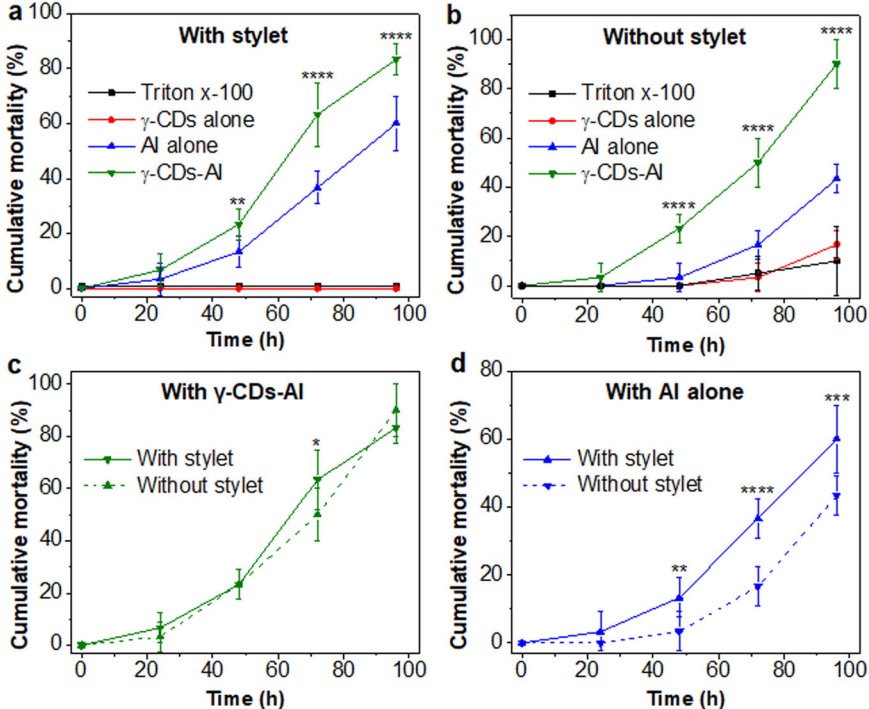

**Fig. 6 | Efficacy of nano formulation on *N. viridula* mortality. a** Cumulative mortality of *N. viridula* caused by AI delivered in γ-CDs compared to AI alone, γ-CDs alone, and 0.1% triton-x 100 alone. **b** Toxicity of AI delivered by γ-CDs compared to AI alone and γ-CDs alone on bugs after removing their stylets. Asterisks in (**a**) and (**b**) indicate significant differences between insect mortality caused by γ-CDs-AI and AI alone (one-way ANOVA with Tukey's post hoc tests, **$**P < 0.001$ and ****$P < 0.0001$). **c** Comparative analysis of stink bug mortality caused by γ-CDs-AI on whole insects and insects after removal of their stylets. **d** Comparative analysis of stink bug mortality caused by AI alone on whole insects and insects after removal of their stylets. Each treatment in panel **a**–**d**, contained 10 bugs on 10 leaves in separate petri dishes and the treatments were replicated 3 times. Asterisks in (**c**) (*$P = 0.0129$, two-tailed unpaired t-test) and (**d**) (**$P = 0.0011$, ****$P < 0.0001$, ***$P = 0.0002$, two-tailed unpaired t-test) indicate significant differences in insect mortality caused by each treatment group. Values in panel **a**–**d** represent means and error bars indicate standard deviation ($n = 3$ biological independent replicates).

was prepared by drop-casting a 5 μL sample (10 μg mL$^{-1}$) onto carbon-coated Cu grids. The height profile of GdCDs and γ-GdCDs were recorded using a tapping mode of atomic force microscopy (AFM) (Dimension 5000; Veeco, USA). To prepare the AFM samples, a silicon wafer first was washed by dipping it in ethanol and acetone for 30 sec each, using a bath sonicator. Subsequently, 5 μL of the respective GdCDs and γ-GdCDs dispersion (1 μg mL$^{-1}$) was dropped on different silicon wafers and dried overnight. The chemical composition of CDs and GdCDs before and after modification with γ-cyclodextrin was investigated using a Nicolet iS50 Fourier transform-infrared (FTIR) advanced KBr gold spectrometer equipped with a Smart iTR diamond ATR device, and analyzed using OMNIC 9 software. The hydrodynamic size and zeta potential of CDs and GdCDs before and after modification with γ-cyclodextrin were measured using a Malvern Zetasizer Nano ZSP instrument at a temperature of 25 °C and analyzed using ZS Xplorer v 3.00 software. For zeta potential and hydrodynamic size measurements, the respective samples at a concentration of 0.1 mg mL$^{-1}$ were dispersed in 10 mM TES buffer (pH 7) and 0.1% triton x-100, respectively. The optical absorbance and fluorescence spectra of both CDs and GdCDs before and after modification with γ-cyclodextrin were recorded using a UV-vis spectrophotometer (UV-2600, Shimadzu, UVProbe 2.50 software) and fluorescence spectrophotometer (Horiba PTI QM-400, FelixG ver. 4.0.1 software), respectively. The conjugation of γ-cyclodextrin to CDs was characterized using $^{11}$B NMR spectroscopy at 298 K on a Bruker Advance I 600Mz NMR spectrometer equipped with a BBFO smartprobe. The $^{11}$B zgbs pulse sequence is used with a spectral width of 38461 Hz and 256 scans with a relaxation time of 0.5 s. The NMR samples of CDs, γ-CDs, and γ-cyclodextrin (5 mg/mL) were prepared in dimethyl sulfoxide-d$_6$ (Sigma-Aldrich).

## Quantification of γ-cyclodextrin fraction in γ-GdCDs

The quantification of γ-cyclodextrin in γ-GdCDs was carried out by studying the interaction of Nile red dye (Sigma-Aldrich), a fluorescent chemical cargo, with γ-cyclodextrin. The ability of γ-cyclodextrin to form an inclusion complex with several chemical cargoes has been reported in previous studies[92–94]. The interaction of Nile red dye with γ-cyclodextrin was investigated by measuring changes in the fluorescent intensity of Nile red dye in the presence of γ-cyclodextrin. Due to the poor solubility of Nile red dye and γ-cyclodextrin in an aqueous medium, a stock solution of Nile red dye (350 μM) and γ-cyclodextrin (400 μM) was prepared in dimethyl sulfoxide (DMSO) (Alfa-Aesar, 99.9%). A 1 mL reaction mixture (30% DMSO in 10 mM (pH 7) TES buffer) containing a fixed amount of dye (15.7 μM) and varying concentrations of γ-cyclodextrins (0, 1, 2, 4, 8, 12, 16, 24, 32, and 50 μM) was prepared and incubated for 1 h at room temperature with shaking. The fluorescence spectra were measured after transferring 200 μL from each sample to a 96-well plate and exciting the mixture at 585 nm wavelength. Based on these spectra, a calibration curve was generated by fitting the non-linear relationship between changes in Nile red dye fluorescence intensity (I-I$_0$/I$_0$) and different concentrations of γ-cyclodextrin. To quantify the fraction of γ-cyclodextrin in γ-GdCDs, 100 μg of γ-GdCDs was incubated with Nile red dye (15.7 μM) for 1 h at room temperature with shaking. The fluorescence intensity of the mixture then was measured after exciting at 585 nm and was fitted into the calibration curve to determine the percentage of γ-cyclodextrin in γ-GdCDs. OriginPro 8.5 software was used for non-linear fitting analysis. One-phase exponential growth function was used as a non-linear fitting model. The degrees of freedom was 4 and the $R^2 = 0.996$.

### Loading of rhodamine 6G (R6G) dye in γ-GdCDs

To load R6G, a 10 μL portion of a 1 mM aqueous solution of R6G was added to a 1 mL of γ-GdCDs dispersion (0.1 mg mL$^{-1}$) prepared in a 10 mM TES buffer (pH 7). This resulted in a final concentration of R6G at 0.01 mM. The reaction mixture then was reacted by shaking in a microplate shaker at 550 rpm for 3 h, which resulted in the formation of the γ-GdCDs-R6G inclusion complex. The formation of γ-GdCDs-R6G inclusion complex was characterized through absorbance and fluorescence spectroscopy.

### Confocal imaging of γ-GdCDs-R6G nanocarriers in leaves

Soybean leaf samples were imaged using a Zeiss 880 inverted confocal laser scanning microscope (CLSM, Zen blue 3.5 software). The first true leaves (two-week-old plants) were treated with 5 μL of γ-GdCDs-R6G nanocarriers dispersion (prepared in 0.1% triton x-100) and incubated for 3 h at room temperature. The leaves treated with 0.1% triton x-100 served as a control. After incubation, 6-mm leaf disks were cut from the treated leaves with the help of a cork borer and placed in a Carolina observation gel chamber (~1 mm in thickness) created on a microscope slide (Corning 2948-75 × 25). The chamber was filled with perfluorodecalin (PFD, 90%, Acros organics) and sealed with a coverslip (VWR). The CLSM imaging settings were as follows: 20x objective lens; laser intensity = 2%; 355 nm laser for γ-GdCDs excitation, 488 nm for R6G excitation, and 633 nm laser for chloroplast excitation; z-stack section thickness = 2 μm; line average = 4; PMT detection range was set to 400–500 nm for γ-GdCDs, 550–640 nm for R6G, and 650–750 nm for chloroplast autofluorescence. The images were acquired using the Zeiss Zen software and processed with Fiji software package (Image-J v 1.54j). The colocalization of γ-GdCDs and R6G with chloroplasts in equidistantly separated images in confocal image overlays was performed using the Coloc 2 function in Fiji software[44,95]. The correlation between the fluorescence signals was analyzed using Pearson's overlap coefficient analysis[73,74]. At least three soybean plants were used for CLSM analysis from the leaf surface to deep in the mesophyll cells.

### Characterization of *N. viridula* tarsi

The optical properties of adult insect tarsi were characterized using CLSM. To prepare the samples for CLSM, the adult insects were anesthetized using $CO_2$ and their tarsi were carefully removed and mounted on glass slides using glycerol (mounting medium). The CLSM imaging settings were as follows: 10x wet objective; z-stack section thickness = 4 μm; line average = 4; PMT detection gain = 500; laser intensity = 2%; and focal plane pinhole size = 1.5 airy units. The tarsal z-stack section thickness was set to 4 μm for the 10x objective to image the whole tarsi having a ~ 500 μm thickness. The tarsi were illuminated with lasers at three distinct wavelengths, 355 nm, 488 nm, and 561 nm, and the emitted fluorescence was collected between 400–500, 500–600, and 600–700 nm wavelengths, respectively. The photomultiplier tube (PMT) detection gain was set to 500 and the focal plane pinhole size to 1.5 airy units. The images were acquired using the Zeiss Zen software and processed with Fiji software package (Image-J v 1.54j).

The tarsal surface of bugs was characterized using a tabletop scanning electron microscope (SEM) (Hitachi TM4000). In brief, the tarsi of anesthetized adult insects were cut off and washed several times with 1x phosphate buffer (pH 7.0) to remove the surface contamination. The tarsi then were fixed in formalin for 2 h at room temperature and washed again multiple times with 1x phosphate buffer (pH 7.0). Afterward, the samples were dehydrated in a graded series of ethanol (30, 50, 75, and 95%) and sputter coated with gold-palladium. Finally, the samples were observed under the SEM at an accelerating voltage of 15 KV.

### Fluorescent chemical cargo delivery by γ-GdCDs nanocarriers into *N. viridula* tarsi

We investigated γ-GdCDs nanocarriers mediated delivery of fluorescent chemical cargo (Nile red dye) into *N. viridula* tarsi by CLSM analysis. Briefly, 250 μL of nanocarrier (2.2 mg mL$^{-1}$) containing 7 μg of Nile red dye was dispersed in 0.1% triton x-100 and evenly sprayed onto Petri dishes using an airbrush. Subsequently, adult stink bugs were placed into the treated dishes and incubated for 16 h. Following incubation, the stink bugs were anesthetized using $CO_2$ and their tarsi were removed. The tarsi were washed three times with 50% acetone (by dipping the tarsi for 15 s each time) to remove any particles adhering to the tarsal surface. The washed tarsi were embedded in an optimal cutting temperature (OCT) compound and frozen at −20 °C inside a cryostat. The frozen samples were used to make 50 μm tarsal sections from the proximal and distal end of the basitarsus (first tarsal segment (T1)). The sections then were transferred to microscope slides (Corning 2948-75 × 25), mounted with glycerol, covered with coverslips (VWR), and the edges of the coverslip were sealed with nail polish before being observed under an inverted confocal microscope. The CLSM imaging settings were as follows: 40x wet objective; 488 nm laser for tarsi excitation and 561 nm laser for Nile red excitation; z-stack section thickness = 1 μm; line average = 4; PMT detection gain = 500; laser intensity = 2%; focal plane pinhole size = 1.5 airy units; PMT detection range was set 500–600 nm for tarsi autofluorescence and 600–700 nm for Nile red dye. A 1 μm thickness of each z-stack section was selected for collecting 50 z-stack sections that provide a detailed resolution for generating the z-stack videos. The 40x objective was chosen for z-stack imaging due to its significantly higher magnification and numerical aperture (1.2) compared to the 10x objective (0.45). For comparison, 250 μL of only Nile red dye solution (containing 7 μg dye) was applied to the insects in a similar way as the γ-GdCDs nanocarriers treatment mentioned above. The Nile red dye solution was prepared by transferring 0.1 mL from the dye stock solution (2.8 mg mL$^{-1}$, prepared in DMSO) to 9.9 mL of 0.1% triton x-100 solution. The images were acquired using the Zeiss Zen software and processed with Fiji software package (Image-J v 1.54j). The movies were created by saving the processed z-stack in AVI format.

### Uptake of γ-GdCDs nanocarriers in *N. viridula* tarsi

The uptake of γ-GdCDs nanocarriers to *N. viridula* tarsi was evaluated by inductively coupled plasma-optical emission spectrometry (ICP-OES). In brief, 250 μL (2 mg mL$^{-1}$) of the nanocarrier dispersion (prepared in 0.1% triton x-100) was evenly sprayed onto each of 10 Petri dishes using an airbrush. Subsequently, two adult stink bugs were placed in each treated dish and incubated for 16 h. The stink bugs then were collected, and their tarsi were removed after being anesthetized using $CO_2$. All of the collected tarsal samples were mixed and washed three times with 0.01 M HNO$_3$ and molecular-grade water to remove any particles adhering to the tarsal surface. To further validate this method, we performed three additional washes, and the resultant solution was analyzed for Gd content via ICP-OES. The analysis revealed no detectable Gd (0.0 ± 0.0 ppm). The tarsi samples then were digested in a metal-grade acidic mixture containing 1 mL HNO$_3$, 0.4 mL HCl, and 0.1 mL H$_2$O$_2$ using a microwave. The digested samples were diluted to 10 mL using molecular-grade water and analyzed using ICP-OES to detect the Gd signal from γ-GdCDs. The Gd content was quantified by generating a linear calibration curve using different concentrations of Gd standards with a correlation coefficient of 0.99 (Supplementary Fig. 19).

### Uptake of γ-GdCDs via tarsi cuticular canals using SEM-EDX

The uptake of γ-GdCDs via tarsi cuticular canals was determined by detecting their Gd signal using a Scanning Electron Microscope equipped with energy dispersive X-ray analysis (SEM-EDX) (FEI Nova

NanoSEM 450). Briefly, 250 μL (2 mg mL⁻¹) of the nanocarrier dispersion (prepared in 0.1% triton x-100) was evenly sprayed onto Petri dishes using an airbrush. Subsequently, the adult stink bugs were placed in a nanocarriers treated petri dish and incubated overnight. The bugs were then transferred onto soybean leaves placed on the 1% agar plates for 2 days. Next, after anesthetizing the insects using $CO_2$, the tarsi were removed and sputter-coated with gold-palladium for 30 s. The samples were observed under the SEM at an accelerating voltage of 10 kV using the Everhard-Thornley detector (ETD). The tarsi were analyzed for the detection of Gd signal on the cuticular canal and no cuticular canal transects using EDX line scan and X-Max detector (Oxford Instruments)[96].

### *N. viridula* mortality assay

The mortality caused by γ-CDs-AI was evaluated on freshly harvested soybean leaves. For this study, trifoliate soybean leaves with similar biometric parameters were collected from 20–30-day-old soybean plants that were cultivated in the growth chambers (see SI methods). To ensure uniformity, the surface area of all the leaves was measured using a portable area meter (LI-COR Biosciences, LI-3000). The collected leaves were transferred to freshly prepared 1% agar (Fisher Scientific) plates. To prevent microbial growth, the agar media was mixed with 30 ppm of streptomycin (Sigma-Aldrich) and 40 ppm of benzimidazole (Sigma-Aldrich) before pouring. After that, the agar plates with leaves were randomly divided into 4 different groups, with each group consisting of 10 plates (40 total plates per replicate). Each group was subjected to specific treatments, including γ-CDs alone, AI alone, γ-CDs-AI, and 0.1% triton x-100. Prior to spraying, all the treatment samples were mixed with 0.1% triton x-100. The treatments were applied to the leaves by spraying an optimized volume of 0.7 mL using an airbrush for 15 s. For the γ-CDs-AI and AI alone groups, the amount sprayed was equivalent to having 7 μg AI (10 ppm, an optimized concentration). After spraying, the leaves were allowed to air dry and then transferred to fresh agar plates to avoid the direct exposure of insects to AI present on the agar surface not covered with the leaf. Based on leaf area calculations, the sprayed amount of AI on the leaf surface in both the treatments was calculated to be ~3 μg. Subsequently, 1–4 days old adult stink bugs were randomly placed in each agar plate, with each group consisting of 10 insects (composed of 5 males and 5 females). The mortality of the stink bugs in each group was determined every 24 h for a period of 96 h. Three independent replications of all treatments were conducted. It is worth noting that placing the leaf on agar plates not only maintained leaf freshness, but also created moisture on the leaf surfaces, which may serve as a medium for the uptake of nanocarrier present on the leaf surface through the stink bug tarsi. These conditions mimic the open field environment where the soybean leaf surface, particularly in the morning, is often covered with dew[97].

To confirm the delivery of AI mediated by γ-CDs-AI and AI alone to *N. viridula* through their tarsi, a follow-up experiment was conducted in which the stylets of these test insects were carefully removed using scissors[98]. Following the stylet removal, the insects were subjected to the different treatments as mentioned above, and their mortality was observed every 24 h for 96 h.

### Statistical analysis

The data were plotted using Origin software (Origin Pro 8.5 Corporation, U.S.) and analyzed using GraphPad Prism 8.0 software. NMR data were analyzed and plotted using MestReNova ver. 15.0.1 software. Experiments were conducted in triplicates, and the descriptive statistics are presented as the mean and standard error of the mean (SEM). A two-tailed Student t-test was conducted at a 95% confidence level to compare the means and SEM of two independent groups (i.e. Figures 2d, e, 3d–f, 5b, 6c, d). To compare the mean and SEM of one variable across three or more independent groups, a one-way ANOVA (analysis of variance) with Tukey's post hoc tests was performed (i.e. Figure 6a, b). The absence of asterisks signifies a lack of significant difference. Mantel-Cox pairwise comparison was performed between the survival of stink bugs after treatment with different formulations and the significant difference was calculated at a confidence level of 95% using Kaplan-Meier statistics (i.e. Supplementary Fig. 17).

### Reporting summary

Further information on research design is available in the Nature Portfolio Reporting Summary linked to this article.

## Data availability

The data that support the findings of this study are available within the article and its supplementary information files. Source data for Figs. 2b–j, 3a, b, d–f, 4b, 5b-d, 6a–d and Supplementary Figs. 2a–d, 4a, b, 5a, 6a, b, 7, 8, 9a, b, 10, 11, 14, 15, 17, 19 are provided in the source data file. The supplementary file contains a NP synthesis schematic (Supplementary Fig. 1), cost analysis for γ-CDs nanocarrier synthesis (Supplementary Table 1), TEM images of nanocarriers (Supplementary Fig. 2), 11B NMR spectra of CDs, γ-cyclodextrin, and γ-CDs (Supplementary Fig. 3), absorbance and fluorescence spectra of CDs and γ-CDs (Supplementary Fig. 4), absorbance spectra and solubility of Nile red dye in the presence of different percentages of dimethyl sulfoxide (DMSO) (Supplementary Fig. 5), hydrodynamic size and zeta potential of γ-CDs and γ-CDs-AI (Supplementary Fig. 6), FTIR spectrum of γ-CDs alone, AI alone, and γ-CDs-AI (Supplementary Fig. 7), confocal images of interactions between Nile red dye and β-cyclodextrins (Supplementary Fig. 8), absorbance and fluorescence spectra of γ-GdCDs, R6G, and γ-GdCDs-R6G (Supplementary Fig. 9), confocal images of fluorescent chemical cargo (R6G) interactions with leaf surface (Supplementary Fig. 10), leaf surface rainfastness analysis of AI alone and Y-CDs-AI at 5.5 mm rainfall (Supplementary Fig. 11), optical characterization of 50 μm tarsal sections (Supplementary Fig. 12), schematic of fluorescent cargo delivery to stink bugs tarsi by nanocarriers (Supplementary Fig. 13), EDX analysis of Gd on insects cuticular canal (Supplementary Fig. 14), SEM imaging and respective EDX analysis of Gd on hemocytes extracted from insects not treated with nanocarriers (Supplementary Fig. 15), photographs of experimental set-up for insect mortality assays (Supplementary Fig. 16), Kaplan-Meier survival analysis test of stink bugs (Supplementary Fig. 17), photographs of stink bugs with stylet and without stylet (Supplementary Fig. 18), and ICP-OES calibration curve of Gd (Supplementary Fig. 19). Source data are provided with this paper.

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

## Acknowledgements

This work was supported by the BASF Corporation, California Research Alliance (CARA) award No. 88554036/ RBC026593 (J.P.G). Dr. Chow-Yang Lee, Department of Entomology, UCR provided Kaplan-Meier statistical advice.

## Author contributions

J.P.G. conceived the idea with W.X. and H.H., and contributed to experimental design, and data analysis. S.S. synthesized and characterized the nanocarriers, designed and conducted the experiments, analyzed the data, and wrote the original manuscript. S.J. loaded R6G dye into the nanocarriers, measured the quantum yield of CDs and GdCDs, and performed AFM characterization of the GdCDs and γ-GdCDs. B.S. performed the TEM imaging of CDs and analyzed their size distribution. T.M.P. reared the insects, provided input into the design, execution, and analysis of insect assays. H.H., W.X., and E.I. contributed to protocol design and data analysis. Y.M.G. assisted with the stink bugs tarsi preparation for confocal analysis, and data analysis. All authors contributed to writing the manuscript.

## Competing interests

A pending patent entitled "Insecticide Delivery by Nanocarriers" is based on this work. Authors J.P.G., S.S., T.M.P., and Y.G. (University of California, Riverside) and H.H., W.X., and E.I. (BASF) are inventors in this patent. The remaining authors declare no competing interests. Specific aspects of the manuscript covered in the pending patent include compositions and methods for targeted delivery of nanocarriers with insecticides.
