## [Transparent Peer Review file · Nature Communications]

Nanocarrier mediated delivery of insecticides into tarsi enhances stink bug mortality

Corresponding Author: Professor Juan Pablo Giraldo

Version 0:

Reviewer comments:

Reviewer #1

(Remarks to the Author)

This manuscript is noteworthy for its efforts in enhancing the targeted delivery of pesticides to pests through nanocarriers. The nanoscale effects of carbon quantum dots and the host-guest chemistry enhance the potential of this system for field applications in the future. However, I still have concerns about some experimental results and designs, and these issues need to be properly resolved before its acceptance for publication. In addition, the standardization of chart production needs to be improved.

1) General comments

The carbon dots nanocarriers were purified by dialysis membrane overnight, which is cost and tedious. It's better to determine the purity of the as-developed CDS. Furthermore, the cost-benefit analysis of the potential application of this study should be tentatively performed.

2) In lines 71-85.

In this section, the author provided an extensive list of cases illustrating instances where the enhancement of pesticide nanocarrier delivery has been rarely achieved, thereby substantiating the novelty and innovative contributions of this manuscript. However, with the explosion of nanocarrier technology in pesticides, some studies have improved the delivery of insecticide to pests, such as (10.1016/j.pestbp.2022.105317, 10.1007/s10340-022-01485-5 and 10.1007/s10340-019-01157-x). How do the authors assess these research cases, and should they be incorporated into the Introduction? Please explain the fundamental distinctions between these studies and the nanocarriers proposed, particularly in the mechanism of enhanced delivery.

3) In line 101.

Why choose γ -cyclodextrin as one of the delivery carriers? Is there a more scientific basis for material selection? Given the cost sensitivity of agricultural inputs, it's worth noting that γ -cyclodextrin may not be the most economical choice among cyclodextrins.

4) lines 120-121.

For agricultural applications, an undoped γ -CDs nanocarrier with comparable physiochemical properties to γ -GdCD was synthesized. How to confirm the comparable physiochemical properties?

5) In lines 156-167 and the Characterization of nanocarriers.

Additional chemical structural characterization, such as NMR, is essential to demonstrate the successful preparation of nanocarriers and provide more comprehensive insights.

6) In lines 202-205 and lines 473-475.

The applied model, methodology, and the degree of fit should be elucidated in this context.

7) In the Restricted nanocarrier and chemical cargo translocation across the plant leaf surface.

It is advisable to conduct testing of nanoformulation content in the leaves after cleaning the leaf surface. This would increase evidence in confirming that the majority of leaves did not absorb the nanoformulation.

Furthermore, it's worth noting that R6G is utilized as the chemical cargo in this section, and its chemical properties and structure differ from those of pesticides. Therefore, is it possible to guarantee that the pesticide used will not be absorbed by soybean leaves?

8) In the Confocal imaging of γ -GdCDs-R6G nanocarriers in leaves.

In a practical pesticide application environment, the majority of active ingredient losses occur due to factors such as

rainwater erosion. Based on the specific application scenario of this study, it would be valuable to demonstrate the nanoformulation's ability to withstand rain washout. This also enables better delivery of insecticide to pests.

9) In lines 311-312.

Please give the source of specific references.

10) In lines 573-577.

The author mentioned that stink bugs feed through their stylets. Do the insects die faster when their stylets are removed? How can you mitigate this interference? Does this factor affect the current experimental results?

Reviewer #2

(Remarks to the Author)

Manuscript review of Nature Communications "Nanocarrier mediated delivery of insecticides into tarsi enhances insect mortality" by Sharma et al

General Comments- This paper describes the synthesis, characterization and use of cyclodextrin modified carbon dots as "molecular baskets" to deliver insecticide cargo to southern green stink bugs, a significant pest of soybean. Importantly, these insect pests are simply a model species; the strategy developed by the authors is applicable to similar stink bug pests on a wide range of crops. The authors correctly note the highly inefficient nature of pesticide use and delivery, as well as the significant negative environmental impacts associated with this current system. The authors also note the pressing need to increase agricultural production to meet the demands of a growing population and the difficulties of a changing climate; nanoscale precision agriculture indeed holds significant promise as a tool in this space. Specifically, the authors synthesized and characterized Gd-doped and undoped carbon dots that were or were not functionalized with cyclodextrin. The nanocarriers were characterized by TEM, AFM, FTIR, DLS, and UV-vis. The loading efficiency of the cyclodextrin, as well as rhodamine dye, and a proprietary insecticide were determined. Confocal imaging was used to investigate penetration of soybean leaves. After stink bug tarsi characterization, the accumulation of dots/AI through the tarsi was investigated by orthogonal techniques (ICP-OES, CLSM). Last, after AI loading, efficacy against the insect was investigated in comparison to neat AI and other controls. In short, the nanocarriers behaved as hypothesized, increasing uptake and toxicity in the insect but not entering plant leaves in detectable quantities. The topic of this paper is timely, adds some important information to our understanding of the potential use on novel nanoscale insecticides, and is within the scope of Nature Communications. The experimental design is robust, and the work was carefully conducted. The authors conclusions are supported by their data. The paper is well written and clearly presented. However, I have a number of concerns that prevent me from recommending this paper for publication in its current form. With a modest amount of effort, the authors should be able to address these concerns and when that is done, the paper should be acceptable for publication.

Abstract- I think more experimental design details are needed on the leaf and insect assays; as written, the text is too vague to actually figure out what was done (dose, design, controls,...)

Line 15- I'd put a number here; the range is known

Line 19- delete "crops"

Line 20- I'd indicate this was investigated by orthogonal techniques; it's awkward as currently written

Line 21- define the cargo investigated; more details on the soybean leaf assay are needed (the fact that it was only 3 hours is important)

Line 34- quantify the damage

Line 44- again, quantify the inefficiency

Line 54- should be "pathogen"

Line 60- insert "the" before "insect"

Lines 60-65- give quantitative information here

Line 73- not sure how you are defining safety?

Lines 77-80- give quantitative information for "enhanced"

Line 91- give the size of the pores

Line 109- I'd like another sentence of detail on reference 41

Lines 115-122- I'd delete this text; not sure you want results in your introduction. Finish the introduction with an impact or significance statement; explicitly tell the reader why this work is important

Lines 128-133- a lot of this seems like methods, not results?

Lines 137 and 142- statistically significant changes?

Line 246- There are quite a few papers in the literature looking at foliar uptake of nanoparticles as a function of charge and none seem to report complete exclusion based on charge. It is true that the vast majority of these other materials are inorganic, making ion dissolution a significant complication but I still think you need more detail here. Regardless of charge, based on the size of your materials, I would have really expected detectable amounts to make it in through the stomata. In addition, I'm concerned that since you only did a 3-hour exposure, you may be making too much of the fact that nothing was detected inside of the leaves. In reality, these residues could sit there for days. Some more text is needed qualifying your statements here.

Lines 250-284- I think this text can be condensed significantly; this level of detail is not needed in the main paper

Line 307- seems odd to cite carbon dot delivery to plants since your materials didn't do this.

Line 319- this AI information needs to be in the methods. Also, including a known AI in the leaf and insect assays would have been a good idea for a benchmark

Line 332- how does this percentage compare to the conventional formulation?

Line 343- change bug to insect

Line 364- delete "unprecedented"

Conclusions- I'd add some text saying that an LCA is needed to quantify all benefits and costs. Also, some text on potential regulatory issues with carbon dots might be useful. Last, how common are tarsi in other insect pests (non-stink bugs). Is this a general strategy that could be used against all insect pests? If so, what about the flip side regarding concerns for pollinators? Presumably the actual load of insecticide in the environment would decrease but for those non-target insects that are exposed, the local dose could be much more of a problem.

Lines 399-438- some of this text can go to the SI

Line 421- 35 mg is not great yield; why so low? As you note, carbon dot synthesis should be scalable

Line 485- this is a very short time frame to determine plant uptake; justify/qualify your choice here

Line 523- was this method validated? How do you know it worked?

Line 547- QA/QC on the ICP needs to be in the SI. Measures of linearity, precision and accuracy are needed

Line 557- define the AI

Reviewer #3

(Remarks to the Author)

In their manuscript entitled "Nanocarrier mediated delivery of insecticides into tarsi enhances insect mortality" the authors around Sandeep Sharma report on their findings that carbon dot nanocarriers are able to efficiently mediate the tarsal uptake of insecticides into the stink bug *Nezara viridula*.

In the first part of their manuscript, they describe the production and characterization of carbon dot nanocarriers (CD and Gd-doped CD). After determination of nanocarrier size and their optical properties, their ability to carry R6G as a tracer was tested, foremost their interaction with plants.

In the second part, the authors studied the uptake of CD and Gd-doped CD by insects, namely the stink bug *Nezara viridula*, a pest. They show that uptake of Nile red Gd-CD occurs considerably through the tarsi. The authors propose cuticular pores as the routes of penetration.

The presented work is highly interesting. For a clear presentation I suggest the authors to address the following points that I thought of during reading.

1) Efficiency of treatment. To me 25% more efficient that application without Gd-CD seems very low. This is especially relevant considering that in the exposure experiments triton was used for the application of the insecticides with and without Gd-CD. Usually, insecticides are not applied together with triton. Therefore, for a full comparison of the effects, the authors should also apply the insecticide without triton as a control. A second issue concerns the insect species used in this study. As a proof of principle, this is probably fine. However, as stated in the title, this work is about the effects of Gd-CD insecticide complexes on "insects". Therefore, I would expect to see another insect species treated following this protocol. Preferably, this would include honeybees. In the same direction, the assays have been done on one plant species only (soybean). The insect and plant species chosen (with distinct surface lipid composition) in this work may represent a unique combination that yields the 25% of more efficiency. For a robust conclusion, additional species should be tested.

2) The entry route of Gd-CD through pores in the tarsi is actually a hypothesis. Indeed, the authors spend some volume in describing this possibility. This is an issue that accounts for the attractiveness of this work and requires therefore some attention. Since direct evidence for this route is not provided, I propose to microscope intact tarsi in order to visualize uptake through pore canals. In addition, again regarding the low efficiency of 25% of the system, I would find it interesting to know the density of pore canals in the tarsi to approach the question as to whether substantial uptake is possible by this route.

3) As a minor issue, I would like to have some words on the confocal experiments. The tarsal z-stack section thickness was 4 μm for the 10x objective, while for the uptake situation a thickness of 1 μm is given for the 40x objective. A more comparative set up is desirable. The authors should please also give more information on the settings: what laser intensity in %? How were the data compiled to figures?

Version 1:

Reviewer comments:

Reviewer #1

(Remarks to the Author)

The authors have substantially revised the manuscript and addressed all the comments properly. Therefore, the revised version is worthy of publication.

Reviewer #2

(Remarks to the Author)

Nice job responding to my comments

Reviewer #3

(Remarks to the Author)

RESPONSE TO REVIEWERS

Reviewer #1 (Remarks to the Author):

This manuscript is noteworthy for its efforts in enhancing the targeted delivery of pesticides to pests through nanocarriers. The nanoscale effects of carbon quantum dots and the host-guest chemistry enhance the potential of this system for field applications in the future. However, I still have concerns about some experimental results and designs, and these issues need to be properly resolved before its acceptance for publication. In addition, the standardization of chart production needs to be improved.

Response: We thank the referee for recognizing the impact of this manuscript and providing constructive comments which helped us to improve its quality.

1) General comments

The carbon dots nanocarriers were purified by dialysis membrane overnight, which is cost and tedious. It's better to determine the purity of the as-developed CDS. Furthermore, the cost-benefit analysis of the potential application of this study should be tentatively performed.

Response: Small molecular weight and oligomeric impurities and byproducts are generated from the incomplete carbonization of precursors during carbon dot synthesis (Analyst 2024, 149, 1680–1700). These impurities strongly affect the chemical reactions involved in CD nanocarrier synthesis resulting in characterization errors (Chem. Mater. 2018, 30, 1878–1887). This demands

the inevitable need to purify the as-synthesized CDs. Among different methods (centrifugation, dialysis, gel electrophoresis, or column chromatography) used for CDs purifications; dialysis is considered as the most simple and effective purification method (Langmuir 2019, 35, 9115–9132), hence we used this method.

We conducted a cost analysis for our insecticide nanoformulation, considering various components including precursors, chemical reagents and purification materials, and the amount of material needed per hectare or per plant (Supplementary Table 1). Considering an application rate of 50 g of insecticide per hectare and a 14.2% γ -CDs loading capacity, the total cost of the nanocarriers amounts needed is calculated to be approximately \$128 per hectare (\$0.0004 per plant) in a field having 325,000 crop plants with a recommended 12-inch row spacing. Adding purification costs using a dialysis membrane, increases the nanocarrier cost to ~\$580 per hectare. However, dialysis membranes can be reused and diafiltration large-scale purification techniques (widely used in industry) can replace dialysis. Significant cost reductions of perhaps an order of magnitude could be achieved by purchasing materials in bulk and performing large-scale synthesis.

We added the following text to the MS with this cost analysis:

“The CD nanocarrier synthesis and purification methods were selected to minimize, to the extent possible, the number of steps and reagents needed thereby reducing material and labor costs. We estimated the cost for our insecticide nanoformulation, considering various components including precursors, chemical reagents and purification materials, using market prices and the amount of material needed per hectare or per plant (SI Table 1). Significant cost reductions of

perhaps an order of magnitude could be feasible by purchasing the chemicals in bulk and large-scale synthesis.”

Material	Market price	Company	Website	γ -CDs nanocarrier synthesis per hectare
Urea	\$3.9 per Kg	Duda Energy LLC	https://dudadiesel.com/choose_item.php?id=urea45	\$40.6
Citric acid	\$6.6 per Kg	Duda Energy LLC	https://dudadiesel.com/choose_item.php?id=50ca	\$40.16
3-(3-Dimethylamino propyl)-1-ethyl-carbodiimide hydrochloride (EDC)	\$352.8 per Kg	Chemimpex.com	https://www.chemimpex.com/3-3-dimethylaminopropyl-1-ethyl-carbodiimide-hydrochloride	\$24.7
N-Hydroxysuccinimide (NHS)	\$39 per Kg	Chemimpex.com	https://www.chemimpex.com/n-hydroxysuccinimide	\$1.95
γ -cyclodextrin	\$397.4 per Kg	Chemimpex.com	https://www.chemimpex.com/category/search/cyclodextrins/2691	\$20.4
Subtotal (Precursors and reagents)				\$128
Dialysis membrane	\$13.4 per m	AliExpress.com	https://www.aliexpress.us/item/3256802665108380.html?	\$450
Total (including purification material)				\$578

Supplementary Table 1. Cost analysis for γ -CDs nanocarrier synthesis. Estimated cost of precursors and chemical reagents for nanocarrier synthesis is approximately ~\$130 per hectare in a field having

325,000 crop plants (\$0.0004 per plant) with a recommended 12-inch row spacing. Adding purification costs using a dialysis membrane, increases the nanocarrier cost to ~\$580 per hectare. However, dialysis membranes can be reused and diafiltration large-scale purification techniques (widely used in industry) can replace dialysis. Significant cost reductions of perhaps an order of magnitude could be achieved by purchasing materials in bulk and performing large-scale synthesis.

2) In lines 71-85.

In this section, the author provided an extensive list of cases illustrating instances where the enhancement of pesticide nanocarrier delivery has been rarely achieved, thereby substantiating the novelty and innovative contributions of this manuscript. However, with the explosion of nanocarrier technology in pesticides, some studies have improved the delivery of insecticide to pests, such as (10.1016/j.pestbp.2022.105317, 10.1007/s10340-022-01485-5 and 10.1007/s10340-019-01157-x). How do the authors assess these research cases, and should they be incorporated into the Introduction? Please explain the fundamental distinctions between these studies and the nanocarriers proposed, particularly in the mechanism of enhanced delivery.

Response: We thank the reviewer for suggesting to compare our study with the suggested references. There are significant differences between these studies and our study.

10.1016/j.pestbp.2022.105317: This study solely demonstrated the uptake of cell-penetrating peptide (CPP) in cell lines and isolated gut cells of *Spodoptera frugiperda*, a chewing pest, without enhancement in insecticide delivery to pests. In contrast, our study revealed CDs uptake through the tarsi of live insects and enhanced pesticide delivery efficacy.

10.1007/s10340-022-01485-5 and 10.1007/s10340-019-01157-x: In these studies, the authors showcase the utilization of star polycation nanoparticles (NPs) for delivering dsRNA-based pesticides to manage aphids. The NPs are observed to traverse the aphids' cuticle following application on the notum (dorsal exoskeleton of the thorax), thereby amplifying the efficacy of dsRNA in aphid control. However, the notum does not directly contact the leaf surface treated with NPs when aphids move on the leaf. Our study diverges significantly, as we demonstrate that nanocarriers remain on the leaf surface, and enter into the insects through pores and glands in their tarsi when in contact with leaves.

The 10.1016/j.pestbp.2022.105317 study did not show any experiments demonstrating NP penetration via the cuticle of live insects. However, the first two studies discussed above have shown significant improvement in the efficacy of RNA based pesticides in sucking insects. Therefore, we highlighted these studies in the introduction as follows:

“Although, some studies demonstrated the penetration of ENMs across the aphid cuticle, enhancing the uptake of RNA-based insecticides^{31,32}, these studies delivered the insecticide through the notum (dorsal exoskeleton of the thorax) which is not in direct contact with the leaf surface as is the insect distal leg segments (tarsi)”

3) In line 101.

Why choose γ -cyclodextrin as one of the delivery carriers? Is there a more scientific basis for material selection?

Given the cost sensitivity of agricultural inputs, it's worth noting that γ -cyclodextrin may not be the most economical choice among cyclodextrins.

Response: The selection of γ -cyclodextrin over other cyclodextrins (α and β) was based on the larger cavity size of γ -cyclodextrin that allows the loading of the Nile red dye and insecticide AI. Minimal host-guest interactions were observed between the Nile red dye (318.37 g/mol) and the smaller cavity sized β -cyclodextrins (Supplementary Fig. 8). Furthermore, the loading capacity of AI (500 g/mol) in β -cyclodextrin CDs was determined to be only $1.4 \pm 0.6\%$. Hence, we performed nanocarrier functionalization with larger cavity sized γ -cyclodextrin rather than β -cyclodextrin. In addition, γ -cyclodextrins have been reported to be more biocompatible than the other α and β cyclodextrins because of their lower impact on cellular lipids (J. Biomed. Mater. Res. A 2005, 74, 454–460; Toxicological Comparison of Cyclodextrins, in Proceedings of the Eighth International Symposium on Cyclodextrins 149–155 (Springer Netherlands, 1996)). We have expanded on this topic in the revised manuscript:

“The selection of γ -cyclodextrin over other cyclodextrins (α and β) was based on the larger cavity size of γ -cyclodextrin that allows the loading of the Nile red dye and insecticide AI as described above. Minimal host-guest interactions were observed between the Nile red dye (318.37 g/mol) and the smaller cavity sized β -cyclodextrins (Supplementary Fig. 8). The loading capacity of AI (500 g/mol) in β -cyclodextrin CDs was determined to be only $1.4 \pm 0.6\%$. In addition, γ -cyclodextrins have been reported to be more biocompatible than the other α and β cyclodextrins because of their lower impact on cellular lipids”

SI Figure 8: No interaction detected between Nile red dye and β-cyclodextrins. There were no significant changes in Nile red fluorescence intensity in the presence of different concentrations of β-cyclodextrin. Values represent means and error bars indicate standard deviation ($n = 3$).

4) lines 120-121.

For agricultural applications, an undoped γ-CDs nanocarrier with comparable physicochemical properties to γ-GdCD was synthesized. How to confirm the comparable physicochemical properties?

Response: The comparable physicochemical properties of γ-GdCD and γ-CDs nanocarriers was confirmed by dynamic light scattering (DLS), zeta potential, and FTIR characterization. DLS

analysis indicated a similar hydrodynamic size of 8.0 ± 0.8 nm and 7.6 ± 0.6 nm for γ -GdCDs and γ -CDs, respectively. Zeta potential values were also similar, -15 ± 1.4 mV and -12.6 ± 0.7 mV for γ -GdCDs and γ -CDs, respectively. FTIR analysis of both γ -GdCDs and γ -CDs confirmed successful functionalization with γ cyclodextrins. These results are discussed in the revised manuscript in the “synthesis and characterization of nanocarriers” section.

5) In lines 156-167 and the Characterization of nanocarriers.

Additional chemical structural characterization, such as NMR, is essential to demonstrate the successful preparation of nanocarriers and provide more comprehensive insights.

Response: We thank the reviewer for suggesting additional characterization. We performed boron-NMR of the carbon dots, γ -cyclodextrin, and γ -CDs. As the γ -cyclodextrin is functionalized to the carbon dots via formation of boronic ester bonds with the aid of 4-carboxyphenylboronic acid linker, therefore, we tracked the boron signal from γ -CDs using boron NMR. The NMR spectra of CDs and γ -cyclodextrin does not show any boron signal; whereas an intense peak has appeared in the spectra of γ -CDs, which clearly showed successful preparation of nanocarriers. We added an SI Figure (Supplementary Fig. 3), and the following results were discussed in the revised manuscript as follows:

“Furthermore, boron NMR spectroscopy was conducted to detect the boron signal from boronic ester bonds established between γ -cyclodextrin and CDs in γ -CDs (Supplementary Fig. 3). This analysis revealed a prominent peak in the spectra of γ -CDs that was not observed in unmodified

γ -cyclodextrin or uncoated CDs, validating the successful functionalization of γ -CDs with γ -cyclodextrins.”

SI Figure 3: Boron NMR spectra of CDs, γ -cyclodextrin, and γ -CDs. γ -CDs spectra with an intense boron peak arising from the boronic ester bond formed between CDs and γ -cyclodextrin. No boron peak was observed in the spectra of CDs and γ -cyclodextrin.

6) In lines 202-205 and lines 473-475.

The applied model, methodology, and the degree of fit should be elucidated in this context.

Response: One-phase exponential growth function was used as a non-linear fitting model.

OriginPro 8.5 software was used for the non-linear fitting. The degree of freedom was 4 and $R^2 = 0.996$. This information is added in the main text in “Quantification of γ -cyclodextrin fraction in γ -GdCDs” method section as:

“OriginPro 8.5 software was used for non-linear fitting analysis. One-phase exponential growth function was used as a non-linear fitting model. The degree of freedom was 4 and the $R^2 = 0.996$.”

7) In the Restricted nanocarrier and chemical cargo translocation across the plant leaf surface. It is advisable to conduct testing of nanoformulation content in the leaves after cleaning the leaf surface. This would increase evidence in confirming that the majority of leaves did not absorb the nanoformulation.

Response: We thank the reviewer for raising this point. We conducted the quantification of nanocarriers in soybean leaves after cleaning the leaf surface. We uniformly applied a known amount of γ -GdCDs on the leaf surface. Treated leaves, untreated leaves, shoots and roots were harvested separately, washed, and digested with acids followed by the quantification of γ -GdCDs based on the Gd signal using ICP-OES (see SI for detailed methods). 24 hours after treatment, the amount of γ -GdCDs in whole plants was quantified to be $11.7 \pm 1.6\%$, with $2.4 \pm 0.8\%$ in treated leaves, $4.0 \pm 1.6\%$ in untreated leaves, $4.9 \pm 1.3\%$ in shoots, and $0.4 \pm 0.04\%$ in roots (Fig. 3d). After 48 h, the amount of γ -GdCDs in whole plants was $14.5 \pm 1.0\%$, with $1.9 \pm 0.4\%$ in treated leaves, $6.5 \pm 2.0\%$ in untreated leaves, $5.5 \pm 1.8\%$ in shoots, $0.6 \pm 0.26\%$ in roots. There was no significant difference in nanocarrier translocation at two different time points of 24 and 48 h. This indicates that 2 days after nanoformulation application, over 85% of the nanocarriers remained on the leaf surface.

Figure 3d. Most nanocarriers remain on the leaf surface two days after application of nanoformulation. ICP-OES analysis of γ -GdCDs content in different plant parts after 24 and 48 h of nanoformulation application to treated leaves ($n = 3$) indicate that over 85% of the nanocarriers remained on the leaf surface. Values represent means and error bars indicate standard deviation.

The figure was added to the revised manuscript, and the following results discussed in the main text:

“Translocation analysis of γ -GdCDs from treated leaves to different parts of soybean plants based on ICP-OES indicated that most nanocarriers remain on the leaf surface after two days of exposure. After 24 h of exposure, the amount of γ -GdCDs in whole plants was $11.7 \pm 1.6\%$, with $2.4 \pm 0.8\%$ in treated leaves, $4.0 \pm 1.6\%$ in untreated leaves, $4.9 \pm 1.3\%$ in shoots, and $0.4 \pm 0.04\%$ in roots (Fig. 3d). After 48 h, the amount of γ -GdCDs in whole plants was $14.5 \pm 1.0\%$, similar to 24 h exposure ($P > 0.05$), with $1.9 \pm 0.4\%$ in treated leaves, $6.5 \pm 2.0\%$ in untreated

leaves, $5.5 \pm 1.8\%$ in shoots, $0.6 \pm 0.26\%$ in roots. After two days of application, over 85% of the nanocarriers remained on the leaf surface.”

Furthermore, it's worth noting that R6G is utilized as the chemical cargo in this section, and its chemical properties and structure differ from those of pesticides. Therefore, is it possible to guarantee that the pesticide used will not be absorbed by soybean leaves?

Response: In the revised manuscript, we tested the penetration of insecticide AI using γ -CDs-AI or AI alone in soybean leaves through LC-MS (Fig. 3e), following a previously reported method (see detailed method in SI) (Pest Manag. Sci. 2019, 75, 3405–3412). The cuticular wax on the leaf surface acts as a permeation barrier for the penetration and uptake of active ingredients. Therefore, we extracted the cuticular wax after 24 and 48 h of AI treatment and quantified the amount of AI present on the leaf cuticle. After 24 h, γ -CDs-AI exhibited a notably higher AI amount ($95.8 \pm 5.5\%$) on the leaf cuticular surface compared to AI alone ($70.9 \pm 2.7\%$) ($P < 0.05$). After 48 h, the AI percentage from both γ -CDs-AI and AI alone decreased to $85.5 \pm 2.4\%$ (10% decrease) and $64.6 \pm 5.3\%$ (6% decrease), ($P < 0.05$) respectively. This decrease in AI content by 10% on leaf surface in case of γ -CDs-AI, can be attributed to the release of AI from γ -CDs-AI followed by AI penetration inside the leaf. Nonetheless, the significantly higher AI percentage retained by γ -CDs-AI after both 24 and 48 h confirmed the nanocarrier's superior AI retention ability on the leaf surface compared to AI alone.

Figure 3e: Insecticide AI on the leaf cuticle surface after application using γ -CDs-AI or AI alone. Most insecticide AI remained on the leaf surface when applied as γ -CDs-AI in comparison to AI alone ($n = 3$) (independent t-test, $**P < 0.05$). Values represent means and error bars indicate standard deviation.

The figure was added to the revised manuscript, and the following results discussed in the main text:

“Similarly, most of AI remained on the soybean leaf cuticular surface after being treated either with the AI loaded γ -CDs (γ -CDs-AI) or AI alone, as quantified by LC-MS (Fig. 3e). The cuticular wax on the leaf surface acts as a permeation barrier for the uptake of AI.³ After 24 h of exposure, γ -CDs-AI exhibited notably higher AI levels ($95.8 \pm 5.5\%$) on the leaf cuticular surface compared to AI alone ($70.9 \pm 2.7\%$) ($P < 0.05$). After 48 h, the AI percentage for both γ -CDs-AI and AI alone treatments decreased to $85.5 \pm 2.4\%$ and $64.6 \pm 5.3\%$ ($P < 0.05$), respectively. The decrease in AI content by 10% on the leaf surface for the γ -CDs-AI, may be

attributed to the release of AI from γ -CDs-AI followed by AI penetration inside the leaf. Nonetheless, the significantly higher AI percentage retained by γ -CDs-AI after two days confirmed the nanocarrier's superior AI retention ability on the leaf surface compared to AI alone.”

8) In the Confocal imaging of γ -GdCDs-R6G nanocarriers in leaves.

In a practical pesticide application environment, the majority of active ingredient losses occur due to factors such as rainwater erosion. Based on the specific application scenario of this study, it would be valuable to demonstrate the nanoformulation's ability to withstand rain washout. This also enables better delivery of insecticide to pests.

Response: We agree that rainwater erosion in the agricultural field can be one of the major factors for the pesticides active ingredient losses, posing a challenge for various commercial insecticide formulations. To test rainfastness, we performed rainfastness experiments with 2.5 mm and 5 mm of rainfall and compared the rain resistance efficiency of AI alone and γ -CDs-AI on the soybean leaf surface. With 2.5 mm rainfall, γ -CDs-AI demonstrated significantly higher retention of AI on the leaf surface ($54 \pm 3\%$) compared to AI alone ($38 \pm 5\%$) (Fig. 3f). However, with 5.5 mm rainfall events, over 90% of the AI was lost in both cases (Supplementary Fig. 11). This suggests that γ -CDs-AI can mitigate significant AI loss under light rainfall conditions, but may not be as effective under moderate to heavy rainfall. However, we would like to emphasize that effective stink bug population control can be achieved after only 5-6 days of insecticide treatment, therefore, farmers could deliberately avoid insecticide spraying on rainy

days, as is advised by the industry to farmers when applying some pesticide products. The results and discussion were added to the main text as:

“Rainfall contributes to the loss of insecticide AI in the agricultural fields⁴. Therefore, we investigated the rainfastness efficiency of γ -CDs-AI in comparison to AI alone on soybean leaves under simulated rainfall conditions of 2.5 mm and 5 mm. The γ -CDs-AI demonstrated significantly higher retention of AI on the leaf surface ($54 \pm 3\%$) compared to AI alone ($38 \pm 5\%$) ($P < 0.05$) under 2.5 mm rainfall (Fig. 3f). However, in our simulated 5.5 mm rainfall study, over 90% of the AI was lost for both the nanocarrier and AI alone treatments (Supplementary Fig. 11). The moderate retention of AI by γ -CDs-AI during light rainfall events may be due to the weak interactions of γ -CDs with the leaf cuticle containing polysaccharides and waxes. The $n-\pi^*$ transition of the C=O bond at 334 nm in CDs (see Supplementary Fig. 4a) can form $\pi-\pi$ stacking interactions with the leaf cuticle wax aromatic compounds (phenylpropanoids and polyphenols)^{5,6}. Also, -COOH groups of CDs, and -OH groups of γ -cyclodextrin may interact via H-bonding formation with the exposed -OH groups of leaf cuticle wax aliphatic compounds and cuticle polysaccharides⁶. Although γ -CDs-AI can mitigate AI loss under light rainfall conditions, they may not be as effective under moderate to heavy rainfall.

Figure 3f: Leaf surface rainfastness analysis of AI alone and γ -CDs-AI at 2.5 mm rainfall ($n = 3$) (independent t-test, $**P < 0.05$). Values represent means and error bars indicate standard deviation.

SI Figure 11: Leaf surface rainfastness analysis of AI alone and γ -CDs-AI at 5.5 mm rainfall. Values represent means and error bars indicate standard deviation ($n = 3$).

9) In lines 311-312.

Please give the source of specific references.

Response: The reference was added “Sci. Rep. 2019, 9, 10330”.

10) In lines 573-577.

The author mentioned that stink bugs feed through their stylets. Do the insects die faster when their stylets are removed? How can you mitigate this interference? Does this factor affect the current experimental results?

Response: There was a slight increase in the mortality of control insects following the removal of their stylets as shown in the original figures (Fig 6a-b). Control insects treated with surfactant (0.1% Triton X-100), while possessing their stylets, exhibited no mortality after 72 hours (Fig. 6a), while insects with their stylets removed showed 10% mortality after 72 hours (Fig. 6b). Insects with their stylets treated with AI alone reached 60% mortality while those with stylets removed had 43% mortality, suggesting more uptake of AI through stylet feeding. Insects treated with γ -CDs-AI actually had *slightly higher* mortality without stylets than those who had stylets had been removed (Fig 6c). Overall, γ -CDs-AI caused significantly higher mortality than AI alone treatments, regardless of whether the insects had stylets or not.

Reviewer #2 (Remarks to the Author):

Manuscript review of Nature Communications "Nanocarrier mediated delivery of insecticides into tarsi enhances insect mortality" by Sharma et al

General Comments- This paper describes the synthesis, characterization and use of cyclodextrin modified carbon dots as "molecular baskets" to deliver insecticide cargo to southern green stink bugs, a significant pest of soybean. Importantly, these insect pests are simply a model species; the strategy developed by the authors is applicable to similar stink bug pests on a wide range of crops. The authors correctly note the highly inefficient nature of pesticide use and delivery, as well as the significant negative environmental impacts associated with this current system. The authors also note the pressing need to increase agricultural production to meet the demands of a growing population and the difficulties of a changing climate; nanoscale precision agriculture indeed holds significant promise as a tool in this space. Specifically, the authors synthesized and characterized Gd-doped and undoped carbon dots that were or were not functionalized with cyclodextrin. The nanocarriers were characterized by TEM, AFM, FTIR, DLS, and UV-vis. The loading efficiency of the cyclodextrin, as well as rhodamine dye, and a proprietary insecticide were determined. Confocal imaging was used to investigate penetration of soybean leaves. After stink bug tarsi characterization, the accumulation of dots/AI through the tarsi was investigated by orthogonal techniques (ICP-OES, CLSM). Last, after AI loading, efficacy against the insect was investigated in comparison to neat AI and other controls. In short, the nanocarriers behaved as hypothesized, increasing uptake and toxicity in the insect but not entering plant leaves in detectable quantities. The topic of this paper is timely, adds some important information to our

understanding of the potential use on novel nanoscale insecticides, and is within the scope of Nature Communications. The experimental design is robust, and the work was carefully conducted. The authors conclusions are supported by their data. The paper is well written and clearly presented. However, I have a number of concerns that prevent me from recommending this paper for publication in its current form. With a modest amount of effort, the authors should be able to address these concerns and when that is done, the paper should be acceptable for publication.

Response: We thank the referee for acknowledging the novelty and the quality of this manuscript. Your recommendations were valuable to improve our manuscript.

Abstract- I think more experimental design details are needed on the leaf and insect assays; as written, the text is too vague to actually figure out what was done (dose, design, controls,...)

Response: We are limited to the 200 word limit for Nature Communications. However, we added more experimental details in the abstract within this constraint. The abstract now read as follows:

“Current delivery practices for insecticide active ingredients (AI) are inefficient with only a fraction (1-25%) of applied AIs reaching their intended target. Herein, we developed carbon dot (CD) based nanocarriers with molecular baskets (γ -cyclodextrin) (γ -CDs) that enhanced the delivery of AI into insects (southern green stink bugs, *Nezara viridula* L.) via their tarsal pores. *Nezara viridula* is a polyphagous herbivore that feeds on leguminous plants worldwide and a

primary pest of soybean (*Glycine max* L). Most of the nanocarriers and their AI cargo (>85%) did not enter soybean leaves after two days of exposure, rendering them available to the insects on the leaf surface. The nanocarriers entered stinkbugs through their tarsi, enhancing the delivery of a fluorescent chemical cargo by ~2.6 times. The insecticide AI nanoformulation (γ -CDs-AI) at an optimized AI concentration (10 ppm), was 25% more effective in controlling the stinkbugs than AI alone. Styletectomy experiments indicated that the improved AI efficacy was due to the γ -CDs-AI entering through the insect tarsal pores, consistent with fluorescent chemical cargo assays. Nanocarrier mediated insecticide AI delivery through the insect's tarsi enables a new approach for efficient AI delivery, improved integrated pest management, and a more sustainable agriculture.”

Line 15- I'd put a number here; the range is known

Response: The number is added as 1-25% (Nature nanotechnology 2022, 17, 347-360). The sentence is modified as “Current delivery practices for insecticide active ingredients (AI) are inefficient with only a fraction (1-25%) of applied AIs reaching their intended target (Nature nanotechnology 2022, 17, 347-360)”.

Line 19- delete “crops”

Response: Deleted.

Line 20- I'd indicate this was investigated by orthogonal techniques; it's awkward as currently written

Response: Edited.

Line 21- define the cargo investigated; more details on the soybean leaf assay are needed (the fact that it was only 3 hours is important)

Response: We now report in the abstract of the revised MS that the most of the nanocarriers and insecticide AI cargo (>85%) remains on the leaf surface after two days of exposure. This is based on new ICP-OES analysis of nanocarriers and AI in plants in response to reviewer #1 (see point 7). The abstract sentence is modified as:

“Most of the nanocarriers and their AI cargo (>85%) did not enter soybean leaves after two days of exposure, rendering them available to the insects on the leaf surface”

Line 34- quantify the damage

Response: Quantified. The sentence is modified as “Stink bugs (Hemiptera: Pentatomidae) are a major pest of food crops affecting over 60 different crop varieties worldwide, and causing annual crop losses (\$60 million) that may surpass those caused by other insects”.

Line 44- again, quantify the inefficiency

Response: Quantified. The sentence is modified as:

“Although traditional insecticide formulations have been effective in controlling their population, only a small fraction (1-25%) of applied insecticide AI reach their intended target insects ⁷”

Line 54- should be “pathogen”

Response: Corrected

Line 60- insert “the” before “insect”

Response: Inserted

Lines 60-65- give quantitative information here

Response: Ref. Sci Rep. 2019, 9, 10330 did not perform quantitative analysis. However, ref.

PLoS Negl Trop Dis. 2020, 14, e0008365 performed the quantitative analysis of distribution of polyanhydride microparticles vs nanoparticles across the exterior of mosquito cuticle. The value is added and the sentence is modified as:

“Likewise, polyanhydride ENMs (150 nm) distributed more uniformly and in higher concentrations (~2.2-fold) across the exterior of mosquito cuticle compared to microparticles (~2 µm) and entered the internal organs via migration through the cuticle through sclerite junctions.”

Line 73- not sure how you are defining safety?

Response: We corrected this mistake and clarify this in the sentence as:

“However, the nanoformulation offered advantages by enhancing the wettability and adhesion ability of acetamiprid, while demonstrating no effect on the germination rate of *Vicia faba* seeds”.

Lines 77-80- give quantitative information for “enhanced”

Response: Quantitative information is provided. The sentences are modified as:

“Similarly, zein polymer-based avermectin nanoformulation caused enhanced mortality (27%) of *Plutella xylostella* L. moths by improving the avermectin photostability under UV irradiation, in comparison to avermectin emulsifiable concentrates. Graphene oxide-based chlorpyrifos AI nanoformulation enhanced mortality (35%) of *Pieris rapae* L. larvae compared to chlorpyrifos emulsifiable concentrate due to their high adhesion ability on the leaf surface under simulated rainy conditions.”

Line 91- give the size of the pores

Response: The information about pores size is not provided in the cited reference.

Line 109- I’d like another sentence of detail on reference 41

Response: The detailed sentence is added as:

“Tagging chloroplast targeting peptides to β -cyclodextrin functionalized CDs has been shown to enhance nanocarrier mediated delivery of 6-carboxyfluorescein fluorescent chemical cargoes into plant chloroplasts by 23%.”

Lines 115-122- I'd delete this text; not sure you want results in your introduction. Finish the introduction with an impact or significance statement; explicitly tell the reader why this work is important

Response: We summarized these sentences focusing on the bigger picture, and ending with an impact statement. This information can offer readers a clearer understanding of our methodology and approach in demonstrating the proof of concept. The revised text reads as follows:

“We demonstrated proof of concept of tarsal delivery of nanocarriers with γ -cyclodextrin functionalized Gd^{3+} -doped CDs (γ -GdCD). The uptake of the fluorescent nanocarriers with chemical cargoes through stink bug tarsi and their ability to remain on the plant leaf surface was elucidated by confocal microscopy and elemental analysis. For agricultural applications, an undoped γ -CDs nanocarrier with comparable physiochemical properties to γ -GdCD was loaded with insecticide AI (γ -CDs-AI) to demonstrate the efficacy of γ -CDs-AI nanoformulation increasing stink bug mortality. Nanotechnology mediated insecticide AI delivery approaches can enable efficient AI delivery, improved integrated pest management, and a more sustainable agriculture.”

Lines 128-133- a lot of this seems like methods, not results?

Response: We believe the short explanation provided in those sentences offers valuable insights into the fabrication and components of nanocarriers illustrated in Fig. 2a. By including these sentences, readers can gain an understanding of the nanocarrier structure before delving into the characterization results, thereby facilitating their comprehension of the subsequent findings.

Lines 137 and 142- statistically significant changes?

Response: Yes, the statistically significant difference was measured using a two-tailed Student-t test. We modified these sentences as:

“The size distribution of CDs and GdCDs after modification with γ -cyclodextrin was significantly increased from 5.6 ± 1 to 9.0 ± 1.3 nm and 6.1 ± 1.3 to 9.3 ± 1.5 nm ($P < 0.0001$), respectively.”

“After functionalization with γ -cyclodextrin, γ -GdCDs showed a significant increase in thickness from 3.2 ± 0.9 nm to 5.9 ± 1.7 nm ($P < 0.0001$)”.

Line 246- There are quite a few papers in the literature looking at foliar uptake of nanoparticles as a function of charge and none seem to report complete exclusion based on charge. It is true that the vast majority of these other materials are inorganic, making ion dissolution a significant complication but I still think you need more detail here. Regardless of charge, based on the size

of your materials, I would have really expected detectable amounts to make it in through the stomata. In addition, I'm concerned that since you only did a 3-hour exposure, you may be making too much of the fact that nothing was detected inside of the leaves. In reality, these residues could sit there for days. Some more text is needed qualifying your statements here.

Response: We agree not only charge but also size and hydrophobicity play important roles in the uptake of nanocarriers in the leaves via stomata. Besides these nanoparticle properties, surfactant surface tension also plays a major role in the CDs uptake through stomata (ACS Nano 14, 7970–7986 (2020)), whereas the size, charge, and hydrophilicity of the CDs also contribute to reduced translocation across the cuticle. These reasons were explained in the original manuscript as:

“The nanocarrier localization on the leaf surface could be attributed to cuticle size exclusion limit, surfactant surface tension, or repulsive electrostatic interactions with the cell wall. The leaf cuticle possesses <2 nm hydrophilic pores^{8–10}, which may impose a size exclusion limit and prevent the uptake of ~8 nm size γ -GdCDs nanocarriers. In addition, the high surface tension of triton x-100 could prevent the uptake of γ -GdCDs through stomatal pores into the leaf mesophyll. A CD dispersion in triton x-100 surfactant, with a high surface tension of 30 mN / m, prevented CDs uptake in both dicot and monocot plants; whereas CDs dispersion in Silwet L-77 surfactant, with a low surface tension of 22 mN / m, allowed CD uptake¹¹. Repulsion electrostatic interactions between the negatively charged cell walls and negatively charged γ -GdCDs may also inhibit nanocarrier translocation across the leaf epidermis. Pectin in plant cell walls has a negative charge¹², exhibiting a higher affinity towards positively charged NPs¹³. Furthermore, based on NP-leaf interaction empirical models, NPs with a charge below +15 mV

exhibited less foliar uptake efficiencies into mesophyll tissue ¹¹. This nanocarrier and nanoformulation design carrying loaded AI prevents their uptake into soybean leaves making them readily available to stink bugs on the leaf surface.”

We also considered the referee's concern and measured the uptake of CD nanocarriers in different plant parts after 24 and 48 h of exposure using ICP-OES analysis (see reviewer #1 point 7). We found that even after 2 days of exposure, over 85% of the nanocarriers and AI insecticide cargo remained on the leaf surface.

Lines 250-284- I think this text can be condensed significantly; this level of detail is not needed in the main paper

Response: The original information was condensed as suggested. However, reviewer #3 asked to provide more characterization of the tarsi surface so we had to expand this paragraph at the same time.

Line 307- seems odd to cite carbon dot delivery to plants since your materials didn't do this.

Response: Sentence was deleted.

Line 319- this AI information needs to be in the methods. Also, including a known AI in the leaf and insect assays would have been a good idea for a benchmark

Response: Because this AI information is already provided in the methods we removed them in the results. We understand the referee's suggestion regarding including a known AI as a

benchmark, which could indeed provide valuable comparative data. However, integrating another AI into the study would require an additional six months. Future field trials will use a known insecticide that shares similar physicochemical properties with the insecticides employed in this study. This approach will enable us to establish a more comprehensive benchmark and facilitate a more robust evaluation of our technology in real-time conditions. The sentence now reads:

“For determining the efficacy of nanocarrier mediated delivery of AI in agricultural applications, we synthesized undoped nanocarriers without Gd (γ -CDs) and loaded them with an insecticide AI (γ -CDs-AI) (see methods).”

Line 332- how does this percentage compare to the conventional formulation?

Response: It can indeed be challenging to directly compare the loading percentage of AI in nanocarriers with conventional formulations. The conventional formulation (from BASF) utilized in this study is 10% (wt./v) SC (suspension concentrate), in which the AI is suspended in water with the assistance of various polymers and adjuvants using high shear mixing.

Accordingly, based on AI percent, the loading percentage in SC may be considered as 10%.

However, by changing the percent of polymers and adjuvants, the amount of AI suspension in water can be easily adjusted. Therefore, the term "loading" may not be an accurate word to use for SC.

Line 343- change bug to insect

Response: Changed

Line 364- delete “unprecedented”

Response: Deleted

Conclusions- I'd add some text saying that an LCA is needed to quantify all benefits and costs. Also, some text on potential regulatory issues with carbon dots might be useful. Last, how common are tarsi in other insect pests (non-stink bugs). Is this a general strategy that could be used against all insect pests? If so, what about the flip side regarding concerns for pollinators? Presumably the actual load of insecticide in the environment would decrease but for those non-target insects that are exposed, the local dose could be much more of a problem.

Response: We have given these points about LCA and regulations further consideration and expanded on them in the revised manuscript. Tarsi are common to insects, however their structure and composition vary amongst taxa. We don't know yet if this strategy can be used against all insect pests and therefore per request of reviewer #3 we changed the manuscript title to “Nanocarrier mediated delivery of insecticides into tarsi enhances stink bug mortality”. We think pollinators such as bees may lack or have different pores and pore canals than stink bugs presenting an opportunity to design nanocarriers that selectively target harmful insects while reducing impact on beneficial species. We added the following text to the conclusions section:

"Future research involving the impact of γ -CDs-AI on various immature life stages could expand on the benefits and reduce the costs of nanocarrier mediated delivery of insecticide AI technologies. Field trials are also needed to identify and mitigate risks pertaining to human

health and environmental safety, and to comply with regulations that vary depending on the country of use.”

“The impact of insecticide AI delivery on pollinators such as bees should be addressed. Adult bees lack dermal glandular pores present in stink bugs ¹⁷, presenting an opportunity to design nanocarriers that selectively target harmful insects while having minimal impact on beneficial pollinators.”

We also added the following text to the results section with this techno-economic analysis per request of reviewer #1:

“The CD nanocarrier synthesis and purification methods were selected to minimize, to the extent possible, the number of steps and reagents needed thereby reducing material and labor costs. We estimated the cost for our insecticide nanoformulation, considering various components including precursors, chemical reagents and purification materials, using market prices and the amount of material needed per hectare or per plant (SI Table 1). Significant cost reductions of perhaps an order of magnitude could be feasible by purchasing the chemicals in bulk and large-scale synthesis.”

Lines 399-438- some of this text can go to the SI

Response: Synthesis of Gd³⁺ doped carbon dots and undoped carbon dots was moved to SI.

Line 421- 35 mg is not great yield; why so low? As you note, carbon dot synthesis should be scalable

Response: In this study, we used 1 kDa membrane to purify the CDs. However, after changing the membrane to 0.5 kDa, we found recently that the losses of CDs during purification are reduced, which enhances the yield from 35 mg to 80 mg. We are identifying protocols for scaling up the synthesis even further for potential upcoming field trials.

Line 485- this is a very short time frame to determine plant uptake; justify/qualify your choice here

Response: As explained above, we added long term CDs uptake analysis to the revised manuscript, in which we detected the nanocarrier uptake in leaves up to 2 days after application using ICP-OES analysis. We found that even after 48 hours after spraying, over 85% of the nanocarriers remained on the leaf surface. Please see the ICP-OES data and results provided to reviewer #1 point (7).

Line 523- was this method validated? How do you know it worked?

Response: Yes, the method was validated. After washing the tarsi three times, we also performed three more washings and determined the Gd signal using ICP-OES. In the solution collected from the initial three washings, we were able to detect the Gd signal. However, in the solution collected from the later washings, we were not able to detect any Gd signal, which showed that all of the adhered particles were washed out after three washings.

This information is included in the method section “Uptake of γ -GdCDs nanocarriers in *N. viridula* tarsi” paragraph as:

“To further validate this method, we performed three additional washes, and the resultant solution was analyzed for Gd content via ICP-OES. The analysis revealed no detectable Gd (data not shown).”

Line 547- QA/QC on the ICP needs to be in the SI. Measures of linearity, precision and accuracy are needed

Response: The linearity of the ICP is provided with a correlation coefficient of 0.99 (see Supplementary Fig. 19). The precision and accuracy was checked by running the samples of known concentration several times by the technician.

This information is provided in the method section “Uptake of γ -GdCDs nanocarriers in *N. viridula* tarsi” as:

“The Gd content was quantified by generating a linear calibration curve using different concentrations of Gd standards with a correlation coefficient of 0.99 (Supplementary Fig. 19).”

SI Figure 19: ICP-OES calibration curve for a range of Gd standard concentration measured at 335.047 nm wavelength. There was a linear correlation with a coefficient of 0.99.

Line 557- define the AI

Response: AI means active ingredient, as we indicate in line 14. After this, only the abbreviation “AI” is used throughout the manuscript.

Reviewer #3 (Remarks to the Author):

In their manuscript entitled “Nanocarrier mediated delivery of insecticides into tarsi enhances insect mortality“ the authors around Sandeep Sharma report on their findings that carbon dot nanocarriers are able to efficiently mediate the tarsal uptake of insecticides into the stink bug *Nezara viridula*.

In the first part of their manuscript, they describe the production and characterization of carbon dot nanocarriers (CD and Gd-doped CD). After determination of nanocarrier size and their optical properties, their ability to carry R6G as a tracer was tested, foremost their interaction with plants.

In the second part, the authors studied the uptake of CD and Gd-doped CD by insects, namely the stink bug *Nezara viridula*, a pest. They show that uptake of Nile red Gd-CD occurs considerably through the tarsi. The authors propose cuticular pores as the routes of penetration. The presented work is highly interesting. For a clear presentation I suggest the authors to address the following points that I thought of during reading.

Response: We thank the referee for the positive evaluation of this work and providing constructive comments that allowed us to improve the manuscript.

1) Efficiency of treatment. To me 25% more efficient than application without Gd-CD seems very low. This is especially relevant considering that in the exposure experiments triton was used for the application of the insecticides with and without Gd-CD. Usually, insecticides are not applied together with triton. Therefore, for a full comparison of the effects, the authors should also apply the insecticide without triton as a control.

Response: We appreciate the reviewer's perspective. The author's view including of our industry partners is that 25% increase in efficiency is highly significant given the potential reductions in costs related to application of insecticides, and importantly access to markets in countries that are currently demanding the reduction in pesticide applications to be approved by government agencies (Nature Food, 2023, 4, 746–750). We added a few sentences to the manuscript clarifying this significant step towards more precise delivery of insecticides.

“The “Green Deal” approved by the European commission aims to reduce the use and risk of chemical pesticides by 50% by 2030¹⁸. Thus, a 25% increase in insecticide efficiency is highly significant given the potential reductions in the use and risk of insecticides, and in costs related to insecticide applications in the field.”

Also following the reviewer's suggestion, we conducted insect mortality assay by excluding Triton X-100 from both γ -CDs-AI and AI SC formulations. We observed a decrease in the AI insecticide efficacy of 20% and 10% when Triton X-100 was excluded from γ -CDs-AI and AI alone, respectively (see figure below panel a). This reduction could be attributed to the loss of AI during spraying due to splashing and bouncing behavior, resulting in less AI landing on the leaf surface. Formulations lacking Triton X-100 exhibited non-uniform leaf coverage (see figure below panel b) compared to those containing Triton X-100 (see figure below panel c), potentially leading to decreased AI efficacy. Furthermore, the lack of Triton X-100 resulted in significant γ -CDs-AI nanocarrier aggregation (see figure below panel d). It is likely that large droplets of γ -CDs-AI nanocarriers (see figure below panel b) aggregated even further on solid leaf surfaces after gradual drying. All these confounding factors contribute to the reduction in the efficacy of

formulations without triton x-100. The presence of surfactant and adjuvants is quite common in agrochemical formulations including Triton X-100 (<https://doi.org/10.1201/9781351069502>) because they increase the wettability of AI, and decrease the splashing and bouncing behavior of sprayed droplets on the leaf surface (ACS Appl. Mater. Interfaces 2020, 12, 44, 50126–50134), which are major causes of pesticide losses. Therefore, for a more realistic and meaningful comparison of γ -CDs-AI nanocarriers and AI alone we must use a surfactant in this study.

Fig. a) Cumulative mortality of *N. viridula* caused by γ -CDs-AI compared to AI alone, containing no 0.1% triton x-100 ($n = 3$). **b)** Image of soybean leaf sprayed with γ -CDs-AI containing no triton x-100. **c)** Image of soybean leaf sprayed with γ -CDs-AI containing 0.1% triton x-100. **d)** Hydrodynamic size analysis of γ -CDs-AI measured with and without the presence of 0.1% triton x-100.

A second issue concerns the insect species used in this study. As a proof of principle, this is probably fine. However, as stated in the title, this work is about the effects of Gd-CD insecticide complexes on “insects”. Therefore, I would expect to see another insect species treated following this protocol. Preferably, this would include honeybees. In the same direction, the assays have been done on one plant species only (soybean). The insect and plant species chosen (with distinct surface lipid composition) in this work may represent a unique combination that yields the 25% of more efficiency. For a robust conclusion, additional species should be tested.

Response: This is an excellent comment. However, it could take another year or even more to perform this study for a second plant-insect system. The revised manuscript is the proof of concept that nanoformulations designed to remain on the leaf surface, enhance the efficacy of insecticide active ingredients by a novel route of delivery through the insect tarsi, using stink bugs as an insect system. We revised our manuscript title to “Nanocarrier mediated delivery of insecticides into tarsi enhances stink bug mortality” to clarify the insect investigated in this study.

2) The entry route of Gd-CD through pores in the tarsi is actually a hypothesis. Indeed, the authors spend some volume in describing this possibility. This is an issue that accounts for the attractiveness of this work and therefore requires some attention. Since direct evidence for this route is not provided, I propose to microscope intact tarsi in order to visualize uptake through pore canals.

Response: Thank you for your constructive comment. We attempted to visualize the uptake of nanocarriers through pore canals of intact tarsi using a confocal microscopy. However, due to the

high autofluorescence of intact tarsi at the excitation wavelength (355 nm) used for nanocarriers (see Fig. 4a), we encountered significant challenges in tracking the nanocarriers with confocal imaging. To address this, we report in the revised manuscript the uptake of γ -GdCDs via tarsi pore canals by detecting their Gd signal using a scanning electron microscope equipped with energy dispersive X-ray analysis (SEM-EDX). The following results are discussed in the main text:

“The uptake of γ -GdCDs via tarsi pore canals was assessed by detecting their Gd signal using a scanning electron microscope equipped with energy dispersive X-ray analysis (SEM-EDX). The stink bugs were held overnight in a petri dish that had been sprayed with nanocarriers (2 mg mL^{-1}). Then, they were transferred to soybean leaves placed with the abaxial side down on 1% agar plates for 2 days. The tarsi were analyzed for the presence of Gd on pore canal and no pore canal transects using EDX line scan (Fig. 5d,e). We observed a higher Gd signal on the pore canal transects in comparison to no pore canal counterparts (Fig. 5e). The EDX spectrum indicated a Gd peak (Supplementary Fig. 14a) and significantly higher percent of Gd atomic ratio (0.57 ± 0.12) in the pore canal transect compared to the no pore canals (0.1 ± 0.1) ($P < 0.05$) (Supplementary Fig. 14b and inset in Fig. 5d). To confirm the uptake of nanocarriers through the tarsi cuticle, we also extracted hemocytes from the hemolymph collected after pricking the insect’s forelegs and measured the Gd signal using SEM-EDX (Fig. 5f,g). EDX spectra revealed the presence of Gd signal in hemocytes, confirming the uptake of nanocarriers via the tarsi (Fig. 5g). No Gd signal was detected on the hemocytes of untreated insect tarsi (Supplementary Fig. 15a,b).”

Fig. 5: Nanocarrier uptake and fluorescent chemical cargo delivery to stink bugs tarsi. a) Confocal images of tarsi sections (T1) prepared from proximal and distal ends show the uptake of Nile red dye delivered by γ -GdCDs nanocarriers ($n = 3$). Blue indicates the tarsi autofluorescence (Ex. $\lambda = 488$ nm) and red indicates Nile red dye fluorescence (Ex. $\lambda = 561$ nm) (scale bar = 50 μ m).

b) Average fluorescence intensity of Nile red delivered without and with γ -GdCDs nanocarriers ($n = 3$). c) Uptake of nanocarriers by the stinkbug tarsi was determined by the ICP-OES analysis of Gd from the γ -GdCDs core ($n = 3$). d-e) EDX line scan performed on SEM images indicate higher Gd signal on the pore canal transects than in no pore canals of the insect tarsi surface. Inset in (d) shows a significantly higher percent of Gd atomic ratio in pore canal transect than no pore canals ($n = 3$). f-g) SEM-EDX performed on the hemocytes extracted from hemolymph after pricking insect's forelegs indicate the presence of Gd signal (white arrows), and confirms the uptake of γ -GdCDs via tarsi ($n = 3$). Values represent means and error bars indicate standard deviation. Statistical analysis performed using independent student t-test (** $P < 0.05$).

Supplementary Figure 14. Higher Gd signal from γ -GdCD nanocarriers in tarsi pore canal transects relative to no pore canal. EDX analysis performed on (a) pore canal transects

indicated a Gd peak and a significantly higher percent of Gd atomic ratio (0.57 ± 0.12) than in **(b)** no pore canal transects (0.1 ± 0.1) ($n = 3$, $P < 0.05$).

Supplementary Figure 15. No Gd detected in hemocytes from insects without nanocarriers.

a) SEM image of hemocytes extracted from the hemolymph collected after pricking insect's forelegs (not treated with γ -GdCDs). **b)** EDX analysis performed on the hemocytes of control insects without nanocarriers showed no Gd signal ($n = 3$).

In addition, again regarding the low efficiency of 25% of the system, I would find it interesting to know the density of pore canals in the tarsi to approach the question as to whether substantial uptake is possible by this route.

Response: We performed SEM imaging on stink bug tarsi at different magnifications (2500 - 80000 x) and found that the distribution and size of putative pores and canals varies at different magnifications. These results are reported and discussed in the revised manuscript:

“The ventral surface of T1 at low magnification (2500 x) showed what appears to be micron-size openings of $4.5 \pm 1.6 \mu\text{m}$ in length (Fig. 4b-v). At 4000-8000 x, the cuticle shows what seems to be sub-micron size dermal pores ranging from $0.38 \pm 0.1 \mu\text{m}$ (Fig. 4b-vi) to $0.16 \pm 0.02 \mu\text{m}$ (Fig. 4b-vii). At 5000 x magnification, there is a wide distribution of putative pores and canals with an average length of $770 \pm 280 \text{ nm}$ and a width of $160 \pm 37 \text{ nm}$ (Fig. 4b-viii). At even higher magnifications (80000 x), nano-sized pore canals of $55 \pm 12 \text{ nm}$ in length were observed to be distributed more uniformly (Fig. 4b-ix) at a density of $43.55 \pm 9.20 / \mu\text{m}^2$ ³⁴. These tarsi structures can come in contact with the leaf surface^{83,84} and the putative pores and canals ranging from micron to nanoscale size may act as uptake pathways for CDs nanocarriers and AI through the tarsi.”

Fig. 4: Optical and physical properties of stink bug tarsi. a) Confocal images of *N. viridula* tarsi show autofluorescence after excitation at 355 nm and 488 nm wavelengths, but not at 561 nm excitation wavelength (scale bar = 250 μm). b) SEM images of *N. viridula* tarsi. (i) Ventral view of three tarsal segments (T1-T3) and pretarsus (PR). (ii) Ventral view of PR showing an unguitractor plate (U), basipulvillus (BP), two pulvilli (P), two curved claws (CL), and two parapodia (PA). (iii-ix) High magnification images of pulvilli (iii,iv) and T1 (v-ix). Arrows indicate the putative pore (red) and pore canals (white) on the tarsal surface. Inset in (viii) shows the magnified image of pore canals highlighted in red circles.

3) As a minor issue, I would like to have some words on the confocal experiments. The tarsal z-stack section thickness was 4 μm for the 10x objective, while for the uptake situation a thickness of 1 μm is given for the 40x objective. A more comparative set up is desirable.

Response: The tarsal z-stack section thickness was set to 4 μm for the 10x objective because we imaged the entire tarsi in this case (see Fig. 4a). Since the whole tarsi sample is thick ($\sim 500 \mu\text{m}$), using a z-stack section thickness of 1 μm takes almost an hour to collect all the 500 (1 μm each) z-stack sections. The 10x objective was chosen for this purpose because imaging the whole tarsi ($\sim 1.5 \text{ mm}$ long) is difficult at 40x magnification. For the uptake analysis, we used a z-stack section thickness of 1 μm with the 40x objective, because in this case the samples we imaged were different: 50 μm cryostat sections of the tarsi (see Fig. 5a). Collecting 50 z-stack sections, each 1 μm thick, is less time-consuming and provides more detailed information for generating the z-stack video with a large number of (50) z-sections. The 40x objective was chosen because it offers significantly better resolution for the 50 μm tarsi sections compared to the 10x objective. We have clarified this in the revised methods section:

“A 1 μm thickness of each z-stack section was selected for collecting 50 z-stack sections that provide a detailed resolution for generating the z-stack videos. The 40x objective was chosen for z-stack imaging due to its significantly higher magnification and numerical aperture (1.2) compared to the 10x objective (0.45).”

“The tarsal z-stack section thickness was set to 4 μm for the 10x objective to image the whole tarsi having a $\sim 500 \mu\text{m}$ thickness.”

The authors should please also give more information on the settings: what laser intensity in %?

How were the data compiled to figures?

Response: The laser intensity used for both nanocarrier tracking in leaves and insect tarsi was 2%. The confocal images were acquired using the Zeiss Zen software and data was processed using Fiji software (Image-J). The videos were created by saving the processed z-stack in AVI format. This information is provided in the methods.